ecology

pinniped, Bass Strait, Southern Oscillation Index, Southern Annular Mode, phenology, marine predator

**Author for correspondence:**
Johanna J. Geeson
e-mail: geesonh@deakin.edu.au

# Environmental influences on breeding biology and pup production in Australian fur seals

Johanna J. Geeson[1], Alistair J. Hobday[2], Cassie N. Speakman[1] and John P. Y. Arnould[1]

[1]School of Life and Environmental Sciences, Deakin University, Burwood, Victoria, Australia [2]CSIRO Oceans and Atmosphere, Hobart, Tasmania, Australia

JJG, 0000-0002-5295-8152; AJH, 0000-0002-3194-8326; CNS, 0000-0002-0023-518X; JPYA, 0000-0003-1124-9330

Knowledge of factors affecting a species' breeding biology is crucial to understanding how environmental variability impacts population trajectories and enables predictions on how species may respond to global change. The Australian fur seal (*Arctocephalus pusillus doriferus*, AUFS) represents the largest marine predator biomass in southeastern Australia, an oceanic region experiencing rapid warming that will impact the abundance and distribution of prey. The present study (1997–2020) investigated breeding phenology and pup production in AUFS on Kanowna Island, northern Bass Strait. The pupping period varied by 11 days and the median pupping date by 8 days and were negatively correlated to 1- and 2-year lagged winter zonal winds, respectively, within Bass Strait. While there was no temporal trend over the study period, annual pup production (1386–2574 pups) was negatively correlated to 1-year lagged summer zonal winds in the Bonney Upwelling region and positively correlated to the current-year Southern Oscillation Index (SOI). In addition, a fecundity index (ratio of new-born pups to adult females at the median pupping date) was positively correlated with current-year Southern Annular Mode (SAM) conditions. Periods of positive SOI and positive SAM conditions are forecast to increase in coming decades, suggesting advantageous conditions for the Kanowna Island AUFS population.

## 1. Introduction

While predators play an integral role in regulating ecosystems through top-down processes, they are also impacted by bottom-up processes affecting prey availability [1,2]. Environmental

variability has long been recognized as a driver of shifts in the abundance and distribution of many prey species, and this is being exacerbated by anthropogenic change [3,4]. Prey availability is particularly important for breeding animals due to the increased energetic costs of producing and rearing young [5]. Consequently, environmental variability has the potential to impact the reproductive cycles of top-order predators [6,7]. Knowledge of the environmental factors influencing reproduction in predators, therefore, is crucial for understanding how predator populations, and the prey species they depend upon, may respond to environmental variability.

This environmental variability is exacerbated in marine ecosystems due to their high temporal and spatial variability [8,9]. Consequently, income-breeding marine predators, particularly those that breed on land or ice (e.g. seabirds and seals), are susceptible to local environmental perturbations as their foraging range is limited by access to reliable oceanic prey sources near potential sites for mating and parturition [10]. While capital-breeding species are able to expand their foraging grounds in response to the movement of prey as they provision their offspring using pre-accumulated energy stores post-partum, income-breeding species must periodically return from foraging to provision young, restricting the distance they can travel [11]. Furthermore, for income-breeding species, variability in foraging grounds can impact reproductive success due to the fasting capacity limits of, or predation faced by, offspring [12,13].

Otariids (fur seals and sea lions) are income-breeding species, unique among eutherian mammals in that all species, with the exception of the northern fur seal (*Callorhinus ursinus*) and Antarctic fur seal (*Arctocephalus gazella*), undertake the majority of active gestation with concurrent lactation [14,15]. Adult females congregate annually in colonies (usually in late spring/early summer) to give birth to a single offspring. They then begin lactation while fasting and mate 4–12 days later [16], before commencing a pattern of alternating foraging trips to sea with nursing periods ashore for the next 10–36 months (depending on species) [17]. The fertilized blastocyst remains in embryonic diapause for 3.5–5 months (depending on species) before active gestation commences [18,19] such that adult females experience the greatest nutritional demands of mid-lactation and active gestation during winter months [20] when marine productivity is likely to be seasonally reduced [21]. Hence, the population dynamics of otariid seals may be particularly susceptible to variability in prey resources through its effects on maternal foraging conditions as this impacts both current and immediate future offspring [22,23].

Fur seals have highly synchronous pupping seasons, with the majority of births occurring within a three to seven week period [24]. The timing of birth is largely determined by photoperiod control on the date of blastocyst implantation nine months earlier [25]. However, inter-individual and inter-annual variation in pupping date is probably influenced by maternal condition at the time of implantation and during active gestation [26]. Correspondingly, the synchrony and timing of pupping may be reflective of environmental effects on prey resource conditions during these periods and provide insights into their ecological linkages with predator populations [27–29].

Like most fur seal species, the Australian fur seal (*A. pusillus doriferus*, AUFS) is still recovering from the over-exploitation of the commercial sealing era of the eighteenth and nineteenth centuries [30,31]. Its breeding distribution is largely restricted to 14 islands within Bass Strait, the shallow (60–80 m) continental shelf region between the Australian mainland and Tasmania, and its approaches [32]. At an estimated *ca* 85 500–120 000 individuals, the present population size is considered to still be *ca* 28–47% of its pre-sealing levels [30,33]. However, with mean female and male body mass at 76 kg and 279 kg, respectively [31], the species currently represents the largest marine predator biomass in southeastern Australia.

The post-harvest recovery of the AUFS population has been relatively slow compared with other species around the world [34,35], particularly that of its conspecific, the Cape fur seal (*A. p. pusillus*), which has rebounded to become the most numerous fur seal species at *ca* 1.7 million individuals [36]. It has been suggested that the slow population recovery is a consequence of the low marine primary productivity of Bass Strait [37], an extreme contrast to the highly productive waters of the Benguela Current where Cape fur seals forage [38]. The region is influenced by a milieu of competing oceanographic currents, with the oligotrophic and warm East Australia Current and South Australia Current entering from the east and west, respectively, while the cold Sub-Antarctic Surface Water enters from the south, and the seasonal Bonney Upwelling brings cold, nutrient-rich waters to the surface in the western region in summer [39]. The area is also one of the fastest warming oceanic regions in the world [40], and observed and anticipated oceanographic changes have and will continue to alter the distribution, productivity and abundance of prey species [41,42], resulting in overall change in prey diversity.

There is presently little information on how regional environmental variation influences marine prey species and the breeding biology of predators that depend on them [13,23,43–45]. While previous studies

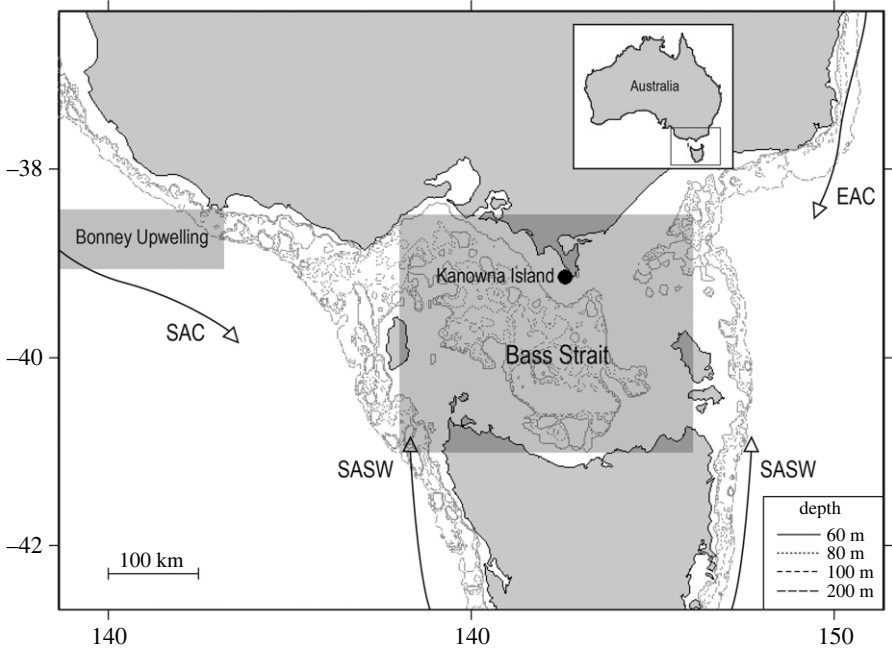

**Figure 1.** Location of the Kanowna Island breeding colony study site (closed circle) within southeastern Australia, the influencing major water bodies and the Bonney Upwelling region. SAC: South Australian Current; EAC: East Australian Current; SASW: Sub-Antarctic Surface Waters. The shaded boxes labelled Bonney Upwelling (BU) and Bass Strait (BS) indicate the region for which local-scale environmental conditions were derived for BU and BS regions, respectively. Inset map shows the position of the region relative to Australia.

found some evidence of AUFS breeding phenology and pup production being influenced by local environmental variables [46], little is known of their relationships with broad-scale climate indices. Environmental information across scales (*sensu* [47]) is necessary for predicting how anticipated climate change impacts will influence the AUFS population and to resolve population differences across colonies [30].

The aims of present study, therefore, were to determine the influence of environmental variation on (i) the timing of breeding and (ii) annual pup production in Australian fur seals. Both of these factors are leading indicators of population growth and abundance, and have relatively short time-to-detect change intervals [48].

# 2. Material and methods

## 2.1. Data collection

The study was conducted between 1997 and 2020 on Kanowna Island (39°09′ S, 146°18′ E), central northern Bass Strait, south-eastern Australia (figure 1). The 32.7 ha granite island hosts the third largest breeding colony of Australian fur seals, with an annual pup production of *ca* 2400–3400 individuals [30,33]. While breeding animals congregate in two main areas (the Main Colony and the East Colony), accounting for greater than 80% of pups born [33], pupping sites are spread around the periphery of the island at low elevations in close proximity to water [49].

Breeding phenology was determined annually from 2003 onwards in the two primary breeding areas. Counts of new-born pups, adult females and territorial males were conducted from an elevated vantage point above each breeding area every 2–3 days (weather permitting) from early November to mid-December. Triplicate counts were conducted by each of 2–3 observers using binoculars and a tally counter at the same time of day (06.00–09.00 AEDT) in each area, to ensure the seals had not yet left the breeding area to thermoregulate at sea [50], and all the counts were then averaged. From these, the pupping period (the number of days covering 90% of all births) and median pupping date were determined [51]. The rate of births followed a normal distribution and sigmoid curves were fitted to the pup counts to calculate both the pupping period and the median pupping date of each year

[46,51]. These breeding phenology parameters (pupping period and median birth date) were selected as they have been shown to be informative metrics in a range of otariid seal studies [13,16,46,51–53], reflecting responses to variation in prey availability and foraging success.

To minimize disturbance to the colony, direct counts of pups were conducted instead of capture–mark–recapture estimates for monitoring annual pup production. While direct counts are known to underestimate capture–mark–recapture estimates in AUFS [33,54,55], previous studies have demonstrated the difference is consistent between years (by a factor of $1.71 \pm 0.04$; [46]) such that they provide an accurate index for assessing trends. Direct counts of all breeding areas were conducted at the end of the pupping period each breeding season. A minimum of three observers searched all areas of the island known to contain pups, counting in predefined sections using binoculars and tally counters from a distance of 10–30 m. Triplicate counts were conducted by each observer and then averaged. In areas with difficult terrain (rock caves, crevasses, etc.), counting was conducted as a single walk-through by each observer and averaged. The total number of pups from the whole-island direct pup count was then used as an index of annual pup production (IAPP).

As fur seals are long-lived mammals (longevity *ca* 21 and 17 years for females and males, respectively [56]) and not all adult females give birth each year [57], fluctuations in annual pup production may not accurately reflect trends in the breeding population size [58]. Re-analysis of the data from a previous study [57] has shown that a majority of adult females that abort their fetus still attend the colony during the breeding season to mate. Consequently, a fecundity index (FI), measured as the ratio of new-born pups to adult females at the median date of pupping (see above), was used to investigate potential environmental influences on the rate of births. These indices were calculated for the main breeding area on Kanowna Island and it was assumed that they were representative of the whole island.

## 2.2. Environmental variables

To investigate environmental influences on breeding biology and pup production in AUFS, separate analyses were conducted on mean seasonal local variables and mean annual broad-scale indices. Local environmental parameters that have been previously shown to influence diet, foraging behaviour and breeding parameters in AUFS [46,59–61] including sea surface temperature (SST), zonal and meridional wind components (*U* and *V* vectors, respectively) and Chlorophyll-*a* (Chl-*a*) (table 1) were investigated. Additionally, geostrophic velocity (GV), a composite of speed and direction of ocean currents, was included in this study due to the potential for changes in ocean surface currents to shift regions of primary productivity [64].

Monthly means of SST, zonal and meridional wind components, Chl-*a*, and GV were obtained for both the Bonney Upwelling and Bass Strait regions (figure 1). Monthly means of SST were derived from CSIRO 3 days composite for the years 1996–2008 (http://www.marine.csiro.au/remotesensing) and from RAMSSA for the years 2009–2020 [68]. Monthly means of Chl-*a* were derived from SeaWiFS and MODIS NASA satellite-based ocean colour imagery for the years 1997–2010 and 2011–2020, respectively [69,70], as SeaWiFS data are only available from September 1997 onwards, analysis with Chl-*a* was only used for the years 1998–2020. In this region, the values for the overlapping period are comparable when these reprocessed products are used. Zonal and meridional wind components and GV were extracted from the NCEP Reanalysis 2 dataset (http://ncdc.noaa.gov). All local environmental variables were extracted at a 4–9 km resolution. Austral winter comprised monthly means from June to August, corresponding to the period of peak milk yield by lactating females [71], and austral summer comprised monthly means from January to March, to encompass the intrusion of nutrient-rich waters from the Bonney Upwelling region, a primary source of Bass Strait productivity [72].

The broad-scale indices assessed included the Southern Oscillation Index (SOI), a major driver of weather in the region associated with changes in sea surface temperature and primary productivity [67], the Southern Annular Mode (SAM), a climate driver that influences rainfall and temperature in Australia [66] and the Indian Ocean Dipole (IOD), a key driver of Australia's climate, based on SSTs of the Indian Ocean [73,74] (table 1). These indices have been found to influence diet [61], foraging effort and foraging success [59,60] in AUFS, as well as breeding parameters in other pinniped species [75,76]. The values for the SOI are one of the key indices for identifying the strength of La Niña and El Niño events [77,78], measuring the monthly mean difference in sea surface pressure from Tahiti to Darwin [79]. Prolonged positive values are associated with La Niña events while prolonged negative values are associated with El Niño events [78,79]. Monthly values were obtained from the National Oceanic and Atmospheric Administration (https://psl.noaa.gov), which were then averaged across all

**5**

**Table 1.** Local-scale environmental variables and broad-scale climate indices used in the analysis to investigate the environmental influences on breeding biology and pup production in Australian fur seals.

| variable | region | temporal scale | abbreviation | description |
|---|---|---|---|---|
| chlorophyll-$a$ | Bonney Upwelling | summer | Chl-$a_{summerBU}$ | an indicator of primary productivity[a] |
| | | 1-year lagged | Chl-$a_{summerBU\ Lag1}$ | |
| | | 2-year lagged | Chl-$a_{summerBU\ Lag2}$ | |
| | | winter | Chl-$a_{winterBU}$ | |
| | | 1-year lagged | Chl-$a_{winterBU\ Lag1}$ | |
| | | 2-year lagged | Chl-$a_{winterBU\ Lag2}$ | |
| | Bass Strait | summer | Chl-$a_{summerBS}$ | |
| | | 1-year lagged | Chl-$a_{summerBS\ Lag1}$ | |
| | | 2-year lagged | Chl-$a_{summerBS\ Lag2}$ | |
| | | winter | Chl-$a_{winterBS}$ | |
| | | 1-year lagged | Chl-$a_{winterBS\ Lag1}$ | |
| | | 2-year lagged | Chl-$a_{winterBS\ Lag2}$ | |
| SST | Bonney Upwelling | summer | SST$_{summerBU}$ | an indicator of the influence of different water bodies moving through the regions[b] |
| | | 1-year lagged | SST$_{summerBU\ Lag1}$ | |
| | | 2-year lagged | SST$_{summerBU\ Lag2}$ | |
| | Bass Strait | winter | SST$_{winterBS}$ | |
| | | 1-year lagged | SST$_{winterBS\ Lag1}$ | |
| | | 2-year lagged | SST$_{winterBS\ Lag2}$ | |

(*Continued.*)

**6**

**Table 1.** (*Continued.*)

| variable | region | temporal scale | abbreviation | description |
|---|---|---|---|---|
| meridional wind component | Bonney Upwelling | summer | wind-$v_{summerBU}$ | primary driver of water flow along the southern coast of Australia[c] |
| | | 1-year lagged | wind-$v_{summerBU}$ Lag1 | |
| | | 2-year lagged | wind-$v_{summerBU}$ Lag2 | |
| | Bass Strait | winter | wind-$v_{winterBS}$ | |
| | | 1-year lagged | wind-$v_{winterBS}$ Lag1 | |
| | | 2-year lagged | wind-$v_{winterBS}$ Lag2 | |
| zonal wind component | Bonney Upwelling | summer | wind-$u_{summerBU}$ | primary source of nutrients for the southern coast of Australia[c] |
| | | 1-year lagged | wind-$u_{summerBU}$ Lag1 | |
| | | 2-year lagged | wind-$u_{summerBU}$ Lag2 | |
| | Bass Strait | winter | wind-$u_{winterBS}$ | |
| | | 1-year lagged | wind-$u_{winterBS}$ Lag1 | |
| | | 2-year lagged | wind-$u_{winterBS}$ Lag2 | |
| meridional GV | Bonney Upwelling | annual | GV-$v$ | a measure of ocean currents that may shift areas of primary productivity in the region[c] |
| | | 1-year lagged | GV-$v$ Lag1 | |
| | | 2-year lagged | GV-$v$ Lag2 | |
| zonal GV | Bonney Upwelling | annual | GV-$u$ | |
| | | 1-year lagged | GV-$u$ Lag1 | |
| | | 2-year lagged | GV-$u$ Lag2 | |
| Indian Ocean Dipole | broad-scale | annual | IOD | a major weather and climate driver with influence across southern Australia[d] |
| | | 1-year lagged | IOD Lag1 | |
| | | 2-year lagged | IOD Lag2 | |

**Table 1.** (*Continued.*)

| variable | region | temporal scale | abbreviation | description |
|---|---|---|---|---|
| Southern Annular Mode | broad-scale | annual | SAM | a major weather and climate driver with influence across Australia[e] |
| | | 1-year lagged | SAM $_{Lag1}$ | |
| | | 2-year lagged | SAM $_{Lag2}$ | |
| Southern Oscillation Index | broad-scale | annual | SOI | a major weather and climate driver with influence across southern Australia and a measure of El Niño and La Niña events[f] |
| | | 1-year lagged | SOI $_{Lag1}$ | |
| | | 2-year lagged | SOI $_{Lag2}$ | |

[a]Thomalla *et al.* [62].
[b]Wijffels *et al.* [63].
[c]Sandery and Kampf [64].
[d]Saji *et al.* [65].
[e]Levenduski and Gruber [66].
[f]Middleton *et al.* [67].

12 months of a calendar year to provide a single value to encompass both the AUFS breeding season and winter pregnancy period.

Previous studies [59,60] have found lags in environmental responses, which may be linked to recruitment or growth periods of important prey species for AUFS [80]. Correspondingly, 1- and 2-year lagged conditions were included in the analyses to investigate the possible impact of lagged conditions on the breeding biology and pup production in AUFS.

## 2.3. Statistical analysis

All statistical analyses were conducted in the R statistical environment (version 4.0.2, R Development Core Team [81]). Data exploration and tests for normality were conducted using the methods described by Zuur *et al.* [82]. Before analyses of environmental variables, correlations between predictor variables at each scale were assessed using Pearson correlation coefficients in the package *corrplot* version 0.84 [83]. Correlations between dependent variables were also determined using this methodology. Where a pair of predictor variables had a strong correlation ($r \geq 0.7$), one variable from that pair was excluded from the candidate set [84]. The variable excluded prior to analysis was selected based on which of the pair would exclude the least number of variables from the variable set.

To avoid overfitting models, variable selection was completed using Random Forests via the *VSURF* package version 1.1.0 [85]. This package was selected as it employs a stepwise procedure for variable selection and the ranking of variable importance using Random Forests [86]. Random Forests were chosen as the method for variable selection because they are useful for minimizing the number of predictor variables from a large set of possibilities [87]. Random Forests were specified with 10 000 trees per forest to provide a robust analysis and a more stable measure of variable importance. Two hundred forests each were grown for the thresholding step and the interpretation step, and 100 forests were grown for the prediction step.

The relationships between the Random Forest selected-dependent variables and the breeding chronology parameters, the IAPP and the FI were each investigated with linear models using the *nlme* package version 3.1–147 [88]. Model selection was conducted using Akaike's information criterion corrected for small sample sizes ($\mathrm{AIC_c}$) [89]. If multiple candidate models (i.e. $\Delta \mathrm{AIC_c} < 4$) were present, a multi-model inference framework with model averaging was adopted to best describe the relationships between the variables [90,91] using the package *MuMIn* v. 1.9.13 [92]. An adjusted $r^2$ was calculated to assess the model fit, and 95% confidence intervals were used to evaluate the significance of each variable. Results are considered significant at the $p < 0.05$ level and data are presented as mean ± standard error unless otherwise stated.

# 3. Results

## 3.1. Timing of breeding

Data were obtained on the timing of breeding for 17 years between 2003 and 2020. The timings of breeding parameters investigated in the present study were significantly, but weakly correlated ($r^2 \leq 0.28$, $p < 0.05$). The pupping period varied from 20 to 31 days ($27 \pm 1$ day), with a shorter period indicating a more synchronous breeding season (electronic supplementary material, table S1). The results of the Random Forest local-scale variable selection for the pupping period included Chl-$a_{\mathrm{winterBU}}$, $\mathrm{SST_{summerBU\ Lag2}}$, $\mathrm{SST_{winterBS\ Lag1}}$, wind-$u_{\mathrm{winterBS\ Lag1}}$ and wind-$u_{\mathrm{winterBS\ Lag2}}$, from which multiple candidate models were derived (table 2). Model averaging of parameter estimates (those variables with 95% confidence intervals excluding 0 [89]; table 3) revealed a significant negative relationship between the pupping period and local environmental variable 2-year lagged winter zonal wind in the Bass Strait region (wind-$u_{\mathrm{winterBS\ Lag2}}$) (figure 2). That is, pupping was more synchronized when these winds were stronger in Bass Strait in the previous 2-year period. While Random Forest variable selection indicated IOD and $\mathrm{SAM_{Lag2}}$ had high levels of importance, the pupping period was not found to be correlated with either of the selected broad-scale indices following model selection (table 3).

The median pupping date was 25 November ± 1 day, with an inter-annual range of 8 days (22–30 November). All Random Forest selected local-scale environmental variables (wind-$u_{\mathrm{winterBS}}$, wind-$u_{\mathrm{winterBS\ Lag1}}$ and wind-$u_{\mathrm{winterBS\ Lag2}}$) each appeared in four of the seven candidate models. While no single model best explained the variation (table 2), model averaging indicated a significant negative relationship between median pupping date and wind-$u_{\mathrm{winterBS\ Lag1}}$ (figure 3*a*). That is, pupping was

**Table 2.** Most-likely models, in descending order, to explain the relationship between variations in pupping period, median pupping date, the IAPP, and the FI and inter-annual variation in environmental conditions. Only models with $\Delta AIC_c < 4$ are shown and only the top 10 of those models are shown. LogLik, log-likelihood of the models; $AIC_c$, selection criteria; $\Delta AIC_c$, the difference between the model's $AIC_c$ and the minimum $AIC_c$ value; $AIC_c$ Wt, weight of Akaike's information criterion corrected for small sample sizes of the models; d.f., degrees of freedom of each model. Additional abbreviations as in table 1.

| candidate models | LogLik | $AIC_c$ | $\Delta AIC_c$ | $AIC_c$ Wt | DF |
|---|---|---|---|---|---|
| *pupping period* | | | | | |
| local-scale variables | | | | | |
| intercept + Chl-$a_{winterBU}$ + SST$_{winterBS\ Lag1}$ + wind-$u_{winterBS\ Lag2}$ | −36.54 | 88.5 | 0.00 | 0.168 | 5 |
| intercept + Chl-$a_{winterBU}$ + wind-$u_{winterBS\ Lag2}$ | −38.68 | 88.7 | 0.16 | 0.155 | 4 |
| intercept + SST$_{winterBS\ Lag1}$ + wind-$u_{winterBS\ Lag2}$ | −39.38 | 90.1 | 1.55 | 0.078 | 4 |
| intercept + SST$_{summerBU\ Lag2}$ | −41.21 | 90.3 | 1.73 | 0.071 | 3 |
| intercept + wind-$u_{winterBS\ Lag2}$ | −41.25 | 90.4 | 1.81 | 0.068 | 3 |
| intercept + SST$_{winterBS\ Lag1}$ | −41.32 | 90.5 | 1.96 | 0.063 | 3 |
| intercept | −42.94 | 90.7 | 2.19 | 0.056 | 2 |
| intercept + Chl-$a_{winterBU}$ + SST$_{summerBU\ Lag2}$ + wind-$u_{winterBS\ Lag2}$ | −37.99 | 91.4 | 2.89 | 0.040 | 5 |
| intercept + SST$_{summerBU\ Lag2}$ + wind-$u_{winterBS\ Lag2}$ | −40.10 | 91.6 | 3.00 | 0.038 | 4 |
| intercept + Chl-$a_{winterBU}$ | −42.10 | 92.0 | 3.50 | 0.029 | 3 |
| broad-scale variables | | | | | |
| intercept + IOD | −41.08 | 90.0 | 0.00 | 0.485 | 3 |
| intercept | −42.94 | 90.7 | 0.72 | 0.339 | 4 |
| intercept + SAM $_{Lag2}$ | −42.76 | 93.4 | 3.36 | 0.091 | 4 |
| intercept + IOD + SAM $_{Lag2}$ | −41.08 | 93.5 | 3.49 | 0.085 | 4 |
| *median pupping date* | | | | | |
| local-scale variables | | | | | |
| intercept + wind-$u_{winterBS\ Lag1}$ | −39.93 | 81.7 | 0.00 | 0.229 | 3 |
| intercept + wind-$u_{winterBS\ Lag1}$ + wind-$u_{winterBS\ Lag2}$ | −35.32 | 82.0 | 0.27 | 0.199 | 4 |
| intercept + wind-$u_{winterBS}$ + wind-$u_{winterBS\ Lag2}$ | −35.57 | 82.5 | 0.77 | 0.156 | 4 |
| intercept + wind-$u_{winterBS}$ + wind-$u_{winterBS\ Lag1}$ | −35.67 | 82.7 | 0.97 | 0.141 | 4 |
| intercept + wind-$u_{winterBS}$ + wind-$u_{winterBS\ Lag1}$ + wind-$u_{winterBS\ Lag2}$ | −33.93 | 83.3 | 1.60 | 0.103 | 5 |
| intercept + wind-$u_{winterBS}$ | −37.90 | 83.7 | 1.94 | 0.086 | 3 |
| intercept + wind-$u_{winterBS\ Lag2}$ | −38.24 | 84.3 | 2.61 | 0.062 | 3 |
| broad-scale variables | | | | | |
| intercept + SOI $_{Lag1}$ | −34.69 | 77.2 | 0.00 | 0.575 | 3 |
| intercept + SOI $_{Lag1}$ + SOI $_{Lag2}$ | −33.84 | 79.0 | 1.80 | 0.234 | 4 |
| intercept + IOD + SOI $_{Lag1}$ | −34.38 | 80.1 | 2.87 | 0.137 | 4 |
| *IAPP* | | | | | |
| local-scale variables | | | | | |
| intercept + wind-$u_{summerBU\ Lag1}$ + Chl-$a_{winterBU}$ | −146.5 | 303.7 | 0.00 | 0.696 | 4 |
| intercept + Chl-$a_{winterBU}$ | −148.9 | 305.3 | 1.66 | 0.304 | 3 |
| broad-scale variables | | | | | |

(*Continued.*)

| candidate models | LogLik | AIC$_c$ | ΔAIC$_c$ | AIC$_c$ Wt | DF |
|---|---|---|---|---|---|
| intercept + SOI + SOI $_{Lag1}$ | −150.1 | 310.7 | 0.00 | 0.297 | 4 |
| intercept + SOI + IOD $_{Lag2}$ | −150.3 | 311.0 | 0.28 | 0.259 | 4 |
| intercept + IOD $_{Lag2}$ + SOI + SOI $_{Lag1}$ | −148.7 | 311.3 | 0.59 | 0.222 | 5 |
| intercept + SOI | −152.1 | 311.7 | 1.01 | 0.180 | 3 |
| *FI* | | | | | |
| local-scale variables | | | | | |
| intercept + Chl-$a_{summerBU}$ + SST$_{summerBU}$ + wind-$u_{summerBU}$ + wind-$v_{summerBU\ Lag1}$ | 32.43 | −43.5 | 0.00 | 0.181 | 6 |
| intercept + Chl-$a_{summerBU}$ + SST$_{summerBU}$ + wind-$u_{summerBU}$ | 29.65 | −43.3 | 0.22 | 0.162 | 5 |
| intercept + Chl-$a_{summerBU}$ + SST$_{summerBU}$ | 27.30 | −43.0 | 0.55 | 0.137 | 4 |
| intercept + Chl-$a_{summerBU}$ + wind-$u_{summerBU}$ | 27.20 | −42.8 | 0.76 | 0.123 | 4 |
| intercept + Chl-$a_{summerBU}$ + SST$_{summerBU}$ + wind-$v_{summerBU\ Lag1}$ | 29.10 | −42.2 | 1.33 | 0.093 | 5 |
| intercept + Chl-$a_{summerBU}$ | 24.71 | −41.4 | 2.11 | 0.063 | 3 |
| intercept + Chl-$a_{summerBU}$ + wind-$u_{summerBU}$ + wind-$v_{summerBU\ Lag1}$ | 28.43 | −40.9 | 2.66 | 0.048 | 5 |
| intercept + Chl-$a_{summerBU}$ + wind-$u_{summerBU}$ + wind-$u_{winterBS}$ | 27.96 | −39.9 | 3.59 | 0.030 | 5 |
| intercept + SST$_{summerBU}$ + wind-$v_{summerBU\ Lag1}$ | 25.64 | −39.7 | 3.87 | 0.026 | 4 |
| broad-scale variables | | | | | |
| intercept + SAM | 24.94 | −41.9 | 0.00 | 0.597 | 3 |
| intercept + SAM + SOI $_{Lag2}$ | 26.20 | −40.8 | 1.12 | 0.341 | 4 |

earlier in the month when zonal winds were stronger in Bass Strait in the previous year's winter period. In analysing broad-scale indices, Random Forest variable selection indicated IOD (one model), SOI $_{Lag1}$ (three models) and SOI $_{Lag2}$ (one model) had high levels of variable importance (table 2). Of these, SOI $_{Lag1}$ was found to have a positive relationship with the median pupping date after model averaging (figure 3*b* and table 3), indicating that positive SOI conditions in the previous year were associated with a later median pupping date.

## 3.2. Pup production

The IAPP varied over the 22 years of the study (1917 ± 67; range: 1386–2574; figure 4), with no significant temporal trend ($r^2 = 0.07$, $p > 0.05$). Random Forest variable selection indicated Chl-$a_{winterBU}$ (two models) and wind-$u_{summerBU\ Lag1}$ (one model) as the most important local-scale environmental variables for modelling. As neither model explained the variation, model averaging was employed (table 2) and a significant negative relationship was found between wind-$u_{summerBU\ Lag1}$ and the IAPP (figure 5*a*). The IAPP was also assessed against broad-scale indices. The results of variable selection showed IOD $_{Lag2}$ (two models), SOI (four models) and SOI $_{Lag1}$ (two models) to have the highest levels of variable importance. While no single model explained the variation, after model averaging, SOI was determined to have a significant positive relationship with the IAPP (figure 5*b*), with the periods of highest pup production occurring during sustained positive SOI values (figure 4). The lagged SOI confidence intervals were skewed away from zero, suggesting that lagged effects were important, but the 95% confidence interval contained zero (table 3).

There was substantial inter-annual variability in the FI (0.41–0.65), with an average of 0.53 ± 0.02 over the 2004–2020 period (figure 6). However, the FI was not significantly related to IAPP ($r^2 < 0.01$, $p > 0.33$).

**Table 3.** The 95% confidence intervals for the environmental influences on pupping period (PP), median pupping date (MPD), the IAPP and the FI based on model averaging. Italicized figures represent variables whose 95% unconditional confidence intervals did not cross zero. Additional abbreviations as in table 1.

| variable | PP | | MPD | | IAPP | | FI | |
|---|---|---|---|---|---|---|---|---|
| | lower | upper | lower | upper | lower | upper | lower | upper |
| **local-scale** | | | | | | | | |
| Chl-$q_{winterBU}$ | −2.58 | 68.79 | | | −842.60 | 5705.00 | | |
| SST$_{winterBS}$ Lag1 | −0.53 | 7.07 | | | | | | |
| wind-$u_{winterBS}$ Lag2 | −3.79 | −0.04 | −2.69 | 0.22 | | | | |
| SST$_{summerBU}$ Lag2 | −0.83 | 4.34 | | | | | | |
| wind-$u_{winterBS}$ Lag1 | | | *−3.38* | *−0.06* | | | | |
| wind-$u_{winterBS}$ | | | −3.08 | 0.28 | | | | |
| wind-$u_{summerBU}$ Lag1 | | | | | *−414.20* | *−3.48* | | |
| wind-$u_{summerBU}$ | | | | | | | −0.075 | 0.01 |
| Chl-$q_{summerBU}$ | | | | | | | *−0.90* | *−0.16* |
| SST$_{summerBU}$ | | | | | | | *0.01* | *0.10* |
| wind-$v_{summerBU}$ Lag1 | | | | | | | −0.01 | 0.12 |
| **broad-scale** | | | | | | | | |
| IOD | −0.79 | 14.44 | | | | | | |
| SAM Lag2 | −2.44 | 4.16 | | | | | | |
| SOI Lag1 | | | *1.14* | *4.30* | −22.79 | 409.40 | | |
| SOI Lag2 | | | −0.73 | 2.63 | | | −0.014 | 0.073 |
| SOI | | | | | *78.89* | *409.40* | | |
| IOD Lag2 | | | | | −116.80 | 1163.00 | | |
| SAM | | | | | | | *0.027* | *0.15* |

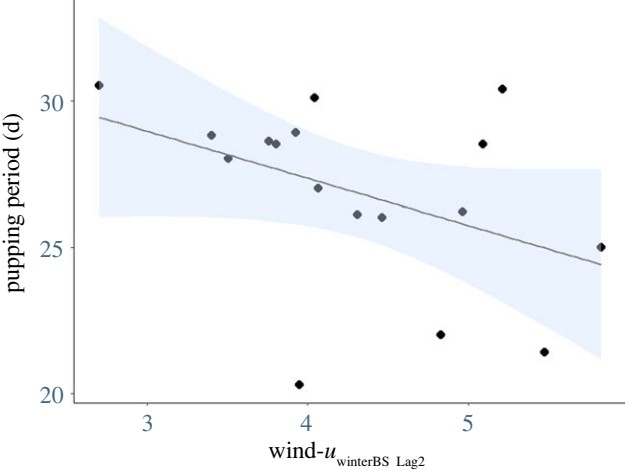

**Figure 2.** The relationship between the pupping period (days) of Australian fur seals and the environmental variable 2-year lagged winter zonal wind in the Bass Strait region (wind-$u_{\text{winterBS Lag2}}$). Blue shading shows the 95% CI.

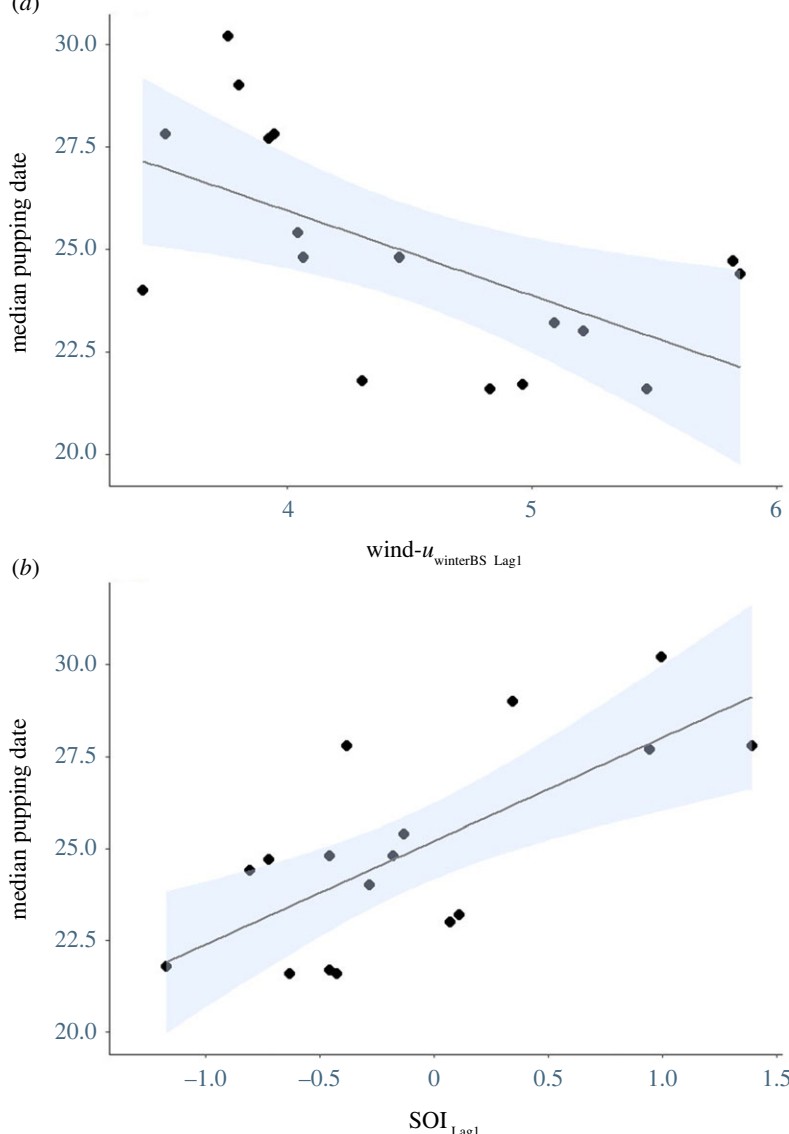

**Figure 3.** Relationships between the median pupping date (days after 31 October) of Australian fur seals at Kanowna Island, northern Bass Strait, and (*a*) 1-year lagged winter zonal wind from the Bass Strait region (Wind-$u_{\text{winterBS Lag1}}$) and (*b*) the 1-year lagged Southern Oscillation Index (SOI $_{\text{Lag1}}$). Blue shading shows the 95% CI.

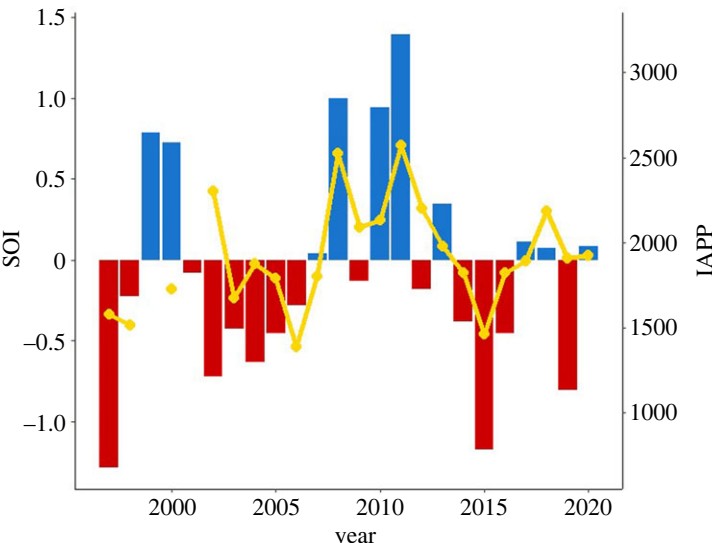

**Figure 4.** Mean annual SOI for the study period 1997–2020 and IAPP for Australian fur seals at Kanowna Island (no data available for 1999 and 2001).

The results of the Random Forest variable selection for the FI and local-scale environmental variables were Chl-$a_{summerBU}$ (eight models), SST$_{summerBU}$ (four models), wind-$u_{summerBU}$ (five models), wind-$u_{winterBS}$ (one model) and wind-$v_{winterBU\ Lag1}$ (four models) (table 2). There were nine candidate models (table 2) and model averaging revealed a significant negative relationship between the FI and Chl-$a_{summerBU}$ (figure 6a) and a significant positive relationship with SST$_{summerBU}$ (figure 6b). Broad-scale indices selected included the SAM (two models) and SOI $_{Lag2}$ (one model), with model averaging indicating only a significant positive relationship between the SAM and the FI (figure 6c and table 3).

# 4. Discussion

In the southeast Australian region, sea surface temperature is predicted to continue warming, leading to systematic changes in surface currents, zonal and meridional wind [93] and, hence, Bass Strait waters are also expected to change [94]. Understanding the environmental factors influencing breeding in the region's marine top-order predators is critical for determining how their populations, and the prey species they depend upon will respond to the predicted change.

The results of the present study have revealed both local-scale variables and broad-scale indices influenced the timing of breeding and pup production in AUFS in northern Bass Strait. While there are potential limitations with this assessment based on assumptions related to the timing of breeding and pup production parameters, the results suggest AUFS are likely to experience further impacts to both their breeding biology and pup production in response to the anticipated oceanographic effects of climate change. The multi-model approach of the present study provides predictions for the direction of change in response to the identified variables. The consistency and explanation for these relationships is strong, but not absolute. Statistical identification of the relationships and the criteria for inclusion mean that additional relationships (e.g. at different lags) may also be important, and it would be easy to suggest that they could be identified with longer time series. Alternatively, rapid change in this region may mean that different processes are operating in the latter period of the time series, and so longer time series will not provide greater clarity. The present study contends that multi-model inference and biological consistency offer the best approach for assessing future outcomes.

## 4.1. Environmental influences on the timing of breeding

The variability in median pupping date and the pupping period observed in the present study can provide insights into the environmental mechanisms influencing breeding in AUFS. Previous studies have shown that larger (presumably older) female otariids give birth later in the pupping season [49,56,95,96] than smaller (presumably younger) females [13,26,57], and that the latter are more likely to abort their fetus in response to decreases in food resources during late pregnancy. Hence, later

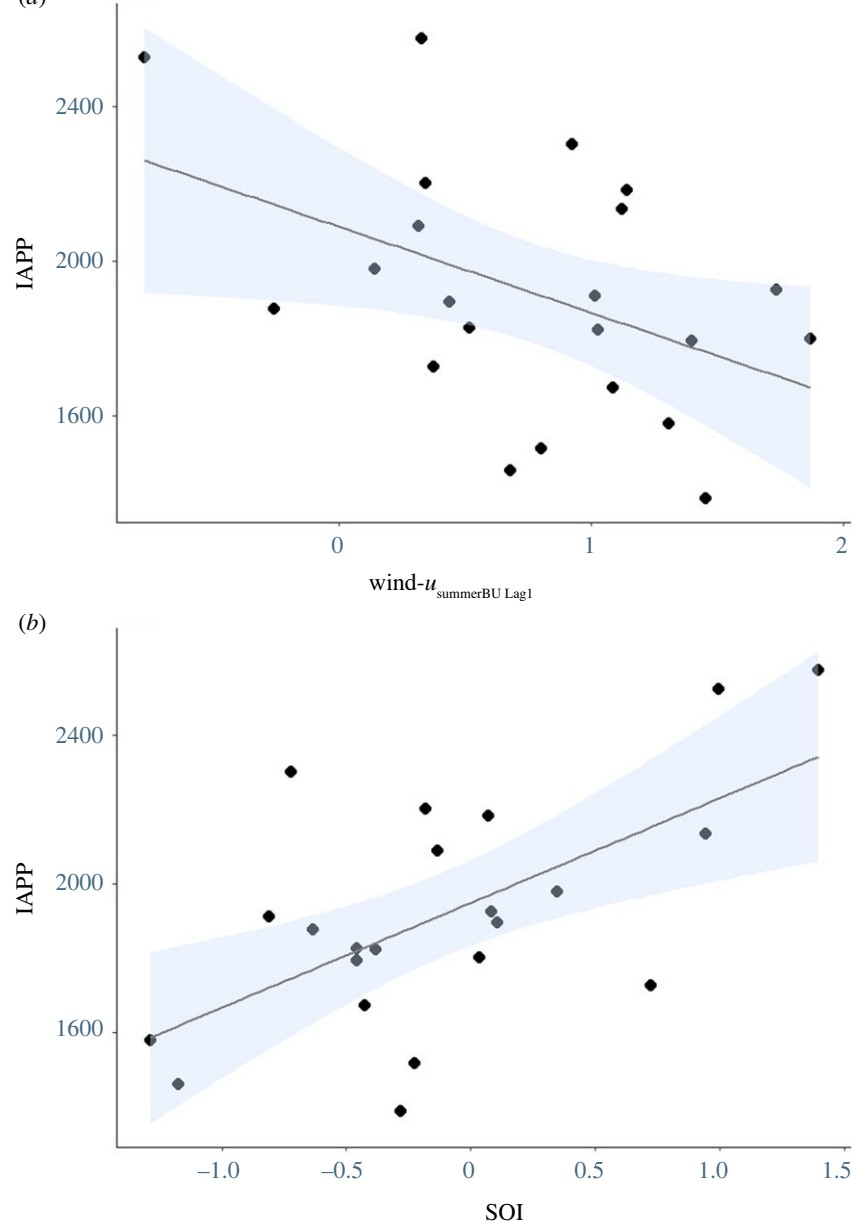

**Figure 5.** Relationships between the IAPP in Australian fur seals at Kanowna Island, northern Bass Strait, and 1-year lagged summer zonal wind in the Bonney Upwelling region (wind-$u_{summerBU\ Lag1}$) (*a*) and the SOI (*b*). Blue shading shows the 95% CI.

median dates of pupping and a greater spread of pupping dates (i.e. longer pupping periods) may indicate poor foraging conditions in the preceding winter, with greater impacts on younger than on older pregnant females.

The present study found a negative relationship between pupping period and winter zonal (westerly) wind in Bass Strait at a lag of 2 years. Lags in environmental responses have previously been observed in foraging ecology studies of AUFS [59,60], which may be linked to recruitment or growth periods of their important prey species. As a primary driver of water flow along the southern coast of Australia, westerly wind has been found to influence primary productivity in Bass Strait [64]. The results of the present study, therefore, suggest increased westerly winds in Bass Strait during winter may improve foraging conditions in subsequent years, with positive consequences for parturition in a greater proportion of younger females. Similarly, the negative relationship between median pupping date and 1-year lagged winter zonal winds in Bass Strait suggests an increased flow of prey into the region in the subsequent year, promoting successful parturition in younger adult females.

Median pupping date was also found to be positively influenced by SOI 1-year lagged. Positive SOI values are associated with warmer surface waters and reduced mixing in southeastern Australia, and

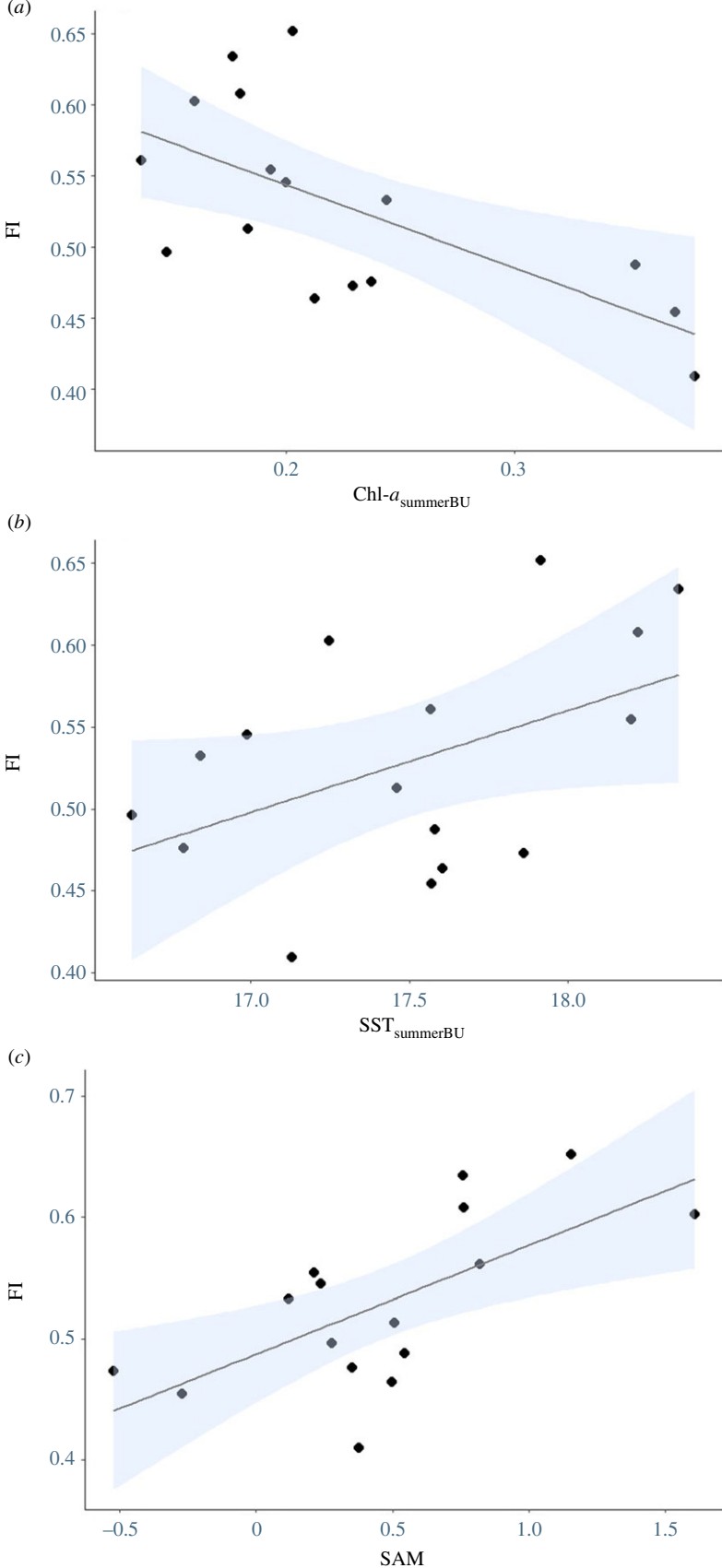

**Figure 6.** Relationships between the FI of Australian fur seals at Kanowna Island, northern Bass Strait, and (*a*) summer Chlorophyll-*a* in the Bonney Upwelling area (Chl-*a*~summerBU~), (*b*) summer SST in the Bonney Upwelling area (SST~summerBU~) and (*c*) the SAM. Blue shading shows the 95% CI.

have been associated with shifts in AUFS prey assemblages [64,97]. The results, therefore, suggest these conditions may negatively impact foraging conditions for younger females, leading to fewer of them pupping in subsequent years. A potential alternative explanation could be related to the finding in the present study that SOI conditions in the current year are positively related to pup production. Previous demographic studies at the nearby Seal Rocks colony (38°31′ S, 146°05′ E) found that first-year survival in AUFS was negatively correlated to cohort size, suggesting a density-dependent effect on maternal provisioning rate through competition for prey resources (Arnould *et al.*, unpublished data), which would probably negatively impact younger females more [98]. Thus, positive SOI conditions in one year could lead to fewer younger females pupping the next and a later median pupping date. Further research is needed to disentangle these apparently complex relationships.

## 4.2. Environmental influences on pup production

The IAPP was found to be negatively influenced by summer zonal winds in the Bonney Upwelling region, 1-year lagged. The seasonally active Bonney Upwelling, driven by Ekman transport and wind forcing along the Bonney Coast in summer, brings large amounts of cold, nutrient-rich water to the surface, leading to high local productivity [99]. These productive waters are entrained into Bass Strait by the easterly flow of the South Australian Current, an extension of the warm Leeuwin Current [64]. A previous study found that lower SST in the Bonney Upwelling region during summer (indicative of greater marine productivity [100]) positively influenced annual AUFS pup production at Kanowna Island in the following breeding season [46]. While SST was not found to be influential on the IAPP in the present study, the influence of 1-year lagged summer easterly winds (i.e. lower zonal winds) in the Bonney Upwelling region is a more direct measure of sub-surface upwelling, indicating increasing marine productivity which may improve prey availability for pregnant females in the subsequent year. Additionally, 1-year lagged zonal wind has been shown to influence both dive and foraging trip duration in AUFS females [60], where stronger westerly winds improved foraging in subsequent years. Therefore, suggesting that zonal winds alter the distribution of prey in AUFS foraging areas. Similarly, foraging effort was found to be negatively influenced by 2-year lagged spring zonal wind [59], highlighting that stronger zonal winds may lead to an increase in future prey abundance in the Bass Strait region.

Of the broad-scale indices investigated in the present study, only SOI in the current year was found to influence pup production in AUFS. In years of sustained positive SOI conditions (figure 4), the IAPP was increased. Conversely, the sustained negative SOI values [101] of the last two decades correspond with the observed IAPP decreases. While the local effects differ, other species of otariids in the Pacific Ocean also respond to changes in the SOI marked by ENSO events. For example, El Niño events cause significant crashes in both the number of pups born and/or high subsequent pup mortality in numerous otariid species [13,102–105]. By contrast, in southeastern Australia, sustained negative SOI conditions are associated with increased upwelling intensity and weakened South Australian and Leeuwin Currents [106,107]. Australia's south coast circulation is affected primarily by zonal winds and the Leeuwin Current [107], and while the enhanced upwelling conditions due to negative phases of the SOI may result in a stronger Bonney Upwelling, promoting higher local marine productivity, a weakened South Australian Current may not move these now productive waters into Bass Strait [72]. Without these productive waters, there are potential effects on prey abundance and distribution within AUFS foraging areas. While the future trends of the SOI remain uncertain, Power & Kociuba [108] suggest that positive SOI conditions will become stronger and more prevalent in the twenty-first century. If this occurs, the results of the present study suggest that the AUFS population at Kanowna Island may experience more frequent advantageous conditions for pup production in the coming years.

The average FI observed in the present study (0.53) is consistent with previous observations of birth rate and late-gestation pregnancy rates in AUFS (53–55%) [57,109]. This low fecundity, and the influence of the generally low primary productivity of the AUFS foraging habitat [37], may be contributing to the slow recovery rate of the population following the eighteenth- and nineteenth-century commercial sealing era [34]. Contrastingly, using necropsies of harvested animals, a previous study reported a fecundity rate of 77.5% in the more abundant conspecific Cape fur seal at a time when its population was rapidly increasing [110]. The results of the present study also revealed substantial inter-annual variation in the FI. As birth rate is used as an important component in calculations of population size in pinnipeds [111,112], this variation suggests population trajectories estimated from short-term pup production trends should be interpreted with caution. Indeed, previous studies have shown that female AUFS that do not successfully complete gestation may continue to suckle their current pup into a second year, which may improve juvenile survival outcomes and influence population estimates [57,113].

The observed inter-annual variation in FI was found to be linked to environmental influences, suggesting that future climate change effects could influence the trajectory of AUFS population through long-term impacts on female pupping rate. However, the mechanisms by which the environment influences pupping rate remain elusive. Surprisingly, the present study found FI was positively influenced by SST and negatively influenced by Chl-*a* concentrations during the preceding summer in the Bonney Upwelling region. However, both lower SST and increased Chl-*a* concentration in the region indicate stronger upwelling events associated with increased productivity and, thus, potentially greater prey availability in the region in subsequent months [99,114]. Furthermore, FI was found to be positively influenced by current-year SAM conditions. A positive SAM index is associated with weaker zonal winds, warm SST anomalies and negative Chl-*a* concentration anomalies in the southeast Australian region [66], leading to lower primary productivity in Bass Strait. Hence, the results of the present study are perplexing and further investigations are required to better understand the linkages between environmental conditions and birth rate in AUFS.

As the IAPP was not correlated with FI, it is possible that pupping rate may be influenced by factors other than prey availability directly. For example, diseases leading to spontaneous abortions could have indirect links to environmental influences. *Brucella sp.* bacteria, a suggested cause of abortion in AUFS and California sea lions (*Zalophus californianus*) [115,116] was found at an increased prevalence in AUFS females during periods of lower than average SST in the Bass Strait region [116]. Furthermore, *Mycoplasma sp.*, a bacteria known to cause abortion if host immunity is decreased, has been found in AUFS [117]. While the prevalence and impact of *Mycoplasma* sp. on pregnant AUFS is not yet known, there could potentially be environmental factors associated with lower SST in Bass Strait impacting their overall health.

In summary, the results of the present study have shown complex interactions between the timing of breeding, fecundity and pup production in AUFS with both local-scale environmental variables and broad-scale climate indices. The predicted future increase in positive SOI trends suggests favourable conditions for pup production may occur more often for AUFS at Kanowna Island, currently the third largest colony for the species. Additionally, the global environment is trending toward a positive SAM as a result of increased warming, increased $CO_2$ concentrations and ozone depletion [94,118]. While global warming conditions are likely to compromise the resilience of many marine species [119], the results of the present study suggest AUFS at Kanowna Island may experience improved reproductive rates. These findings highlight the importance of long-term datasets in detecting environmental influences of relevance for future population management in a changing environment.

Ethics. All research methods were conducted in accordance with the regulations of the Deakin University Animal Ethics committee (Approval A33/2004, A16/2008, A14/2011, B16/2014, B04/2017), Macquarie University Animal Care and Ethics Committee (Approval 97001, 2000/004), University of Melbourne Animal Experimentation Ethics Committee (Approval 01146) and Department of Sustainability and Environment (Victoria, Australia) (wildlife research permits 10000187, 10000706, 10001143, 10001672, 10002269, 10005362, 10007153, 10008286 and 10005848).

Data accessibility. The data are provided in the electronic supplementary material [120].

Authors' contributions. J.J.G.: data curation, formal analysis, investigation, writing—original draft, writing—review and editing; A.J.H.: data curation, formal analysis and writing—review and editing; C.N.S.: data curation, formal analysis and writing—review and editing; J.P.A.: conceptualization, data curation, formal analysis, funding acquisition, investigation, methodology, project administration, resources, supervision, visualization, writing—original draft and writing—review and editing.

All authors gave final approval for publication and agreed to be held accountable for the work performed therein.

Competing interests. The authors declare no competing interests.

Funding. Financial support was provided by research grants from the Australian Research Council, Sea World Research and Rescue Foundation, Winnifred Violet Scott Trust and Holsworth Wildlife Research Endowment.

Acknowledgements. We thank the many staff, students and volunteers involved in the data collection over the years. Logistical support was provided by Parks Victoria, Prom Adventurer Charters (Geoff Boyd) and Best Helicopters (Sean Best and Cameron Lang) and their assistance is gratefully acknowledged. Jason Hartog extracted and provided the environmental data.

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
