## [Peer Review File · Royal Society Open Science]

Review History

RSOS-211399.R0 (Original submission)

Review form: Reviewer 1

Is the manuscript scientifically sound in its present form?

Yes

Are the interpretations and conclusions justified by the results?

Yes

Is the language acceptable?

Yes

Do you have any ethical concerns with this paper?

No

Have you any concerns about statistical analyses in this paper?

No

Recommendation?

Accept with minor revision (please list in comments)

Comments to the Author(s)

The authors examine environmental factors that could be modulating pup production in the Australian fur seal over multiple decades. They do an admirable job of tackling difficult questions, particularly in terms of their consideration of the environmental inputs to consider and their interpretation and discussion of their results. The mechanisms linking predator reproduction to ocean conditions are complex and their manuscript reflects that. Their analyses and results generally seem robust and well thought through. I would suggest that the authors look at streamlining and focusing the introduction somewhat. I recognize that there are a lot of concepts to introduce, both biologically and oceanographically, but some of the information included there felt extraneous or like it could be presented more concisely. I've provided some specific suggestions and comments here:

Lines 51-80: I understand that a lot of this information is useful background, but I believe it could be restructured and rearranged somewhat to introduce the concepts more concisely and smoothly and to transition between the broad introduction and study objectives.

Lines 51-57: This paragraph in particular feels somewhat dropped in. Additionally, the second sentence is true for income breeding pinniped species, but not really for most capital breeders, especially those on the more extreme side of the capital/income axis. Lastly, there is no real explanation of what an income breeder is in this paragraph, which is fine for the audience that studies seabirds/marine mammals, but might be needed clarification for broader audiences (especially since income breeding can mean different things for different organisms).

Lines 73-75: This concept feels like it should be integrated with your earlier description of fur seal reproduction. The statement on line 70-72 ("population dynamics of otariid seals....") summarizes the information in this later paragraph as well as it does that previous paragraph

Lines 107-110: These statements about the broader economic/social implications of your study are interesting, but they are not revisited in the discussion/conclusion of your manuscript. Suggest removing.

Line 135: since you only use the term CMR once, and in that same sentence, I suggest not adding it as an acronym.

Line 354-355: wording suggestion "El Niño events cause significant crashes in both the number of pups born and/or high subsequent pup mortality."

Lines 374-377: It might be interesting to know/discuss how this fecundity rate compares to that of other fur seal species, especially following periods of exploitation.

Review form: Reviewer 2

Is the manuscript scientifically sound in its present form?

No

Are the interpretations and conclusions justified by the results?

No

Is the language acceptable?

Yes

Do you have any ethical concerns with this paper?

No

Have you any concerns about statistical analyses in this paper?

No

Recommendation?

Major revision is needed (please make suggestions in comments)

Comments to the Author(s)

The study has shown a lot of interactions between each of four breeding indicators and a lot of environmental predictors. Firstly, I have some concerns with the reliability of some of the breeding parameters, or the support for them. Actual pup production seems fine - it is a well-established indicator. It is acknowledged (148) that it might not accurately reflect trends in breeding population size, but we are not looking at trends in breeding population size we are looking at environmental influences on breeding biology, at an annual scale (with lags). I am a bit unconvinced about the veracity of the fecundity index introduced next (150): the ratio between females and pups in the colony. It supposes that females that abort their fetuses before their full period still attend the breeding colony in breeding season to mate. The reference given to support this is Gibbens et al. 2010 - but I cannot find any evidence in that paper to support this. A fecundity index such as this would probably need to be assessed by looking at its relationship with pregnancy estimation in females sampled before the breeding season, each year. The median pupping date as an indicator of variability supposes a relationship between median pupping date and abortion rate in younger females (which have been shown to pup earlier than older females), but has such a relationship been shown? Please provide support.

Also, are the breeding parameters related to one another as one would expect? If they are not, this might also put the meaningfulness of some of the relationships with environmental predictors into doubt. For example, the relationship between SOI and pup production suggests that positive SOI has a positive influence on pup production in the same year. But the positive relationship between SOI and median pupping date is taken to mean that positive SOI in one year has a negative influence on foraging in the next year, causing later median pupping date. So it should be expected that later median pupping is associated with lower pup production (same year). Is this the case? Likewise, the relationship of these breeding variables with fecundity, and pupping period, should be established. Is the median date actually positively related to pupping period, as expected (311)? No real evidence is cited to support it. So lack of support for some of these breeding parameters seems a weakness of this study; some seem to be more or less assumed based on reasoning, which makes the study seem highly speculative. I would like to see stronger support for the parameters, or at least greater acknowledgment of their potential limitations and the implications for results and interpretation. And I would like to see how the different breeding indicators used here relate to each other. It feels as though this should be an exercise at the outset, and based on the correlations, or lack thereof between breeding parameters, and/or solid support for them, it might be appropriate to not select one or more of them for modelling with environmental parameters.

More specific comments:

52-57: Not sure that ice (sea ice) should be seen as terrestrial. Suggest changing the sentence as follows: "Consequently, marine predators that breed on land or ice (e.g. pinnipeds, seabirds) are susceptible to local environmental perturbations as their foraging range is limited by access to reliable oceanic prey sources near potential sites for mating and parturition [10]." (However, susceptibility to local environmental perturbations is much more relevant to income breeders

than capital breeders, whose foraging distribution may be much further removed from breeding habitat. Perhaps it should be “Consequently, income breeding marine predators....”, with modification of the next sentence)

87: Is this the average body mass? Please state...

L89: It might be useful to specifically also contrast the slow recovery of the AuFS with the recovery of its conspecific, *A.P. pusillus*

100: For the statement: “There is presently little information on how environmental variation in the region influences marine prey species and the breeding biology of predators that depend on them” - it’s unclear why studies from other regions are cited. E.g. Benguela, South America

216: write out “Two hundred” to start the sentence

255, Fig 2 and elsewhere: based on this, every year in the study is either an El Nino or La Nina year, which is not true, only some of these years would have El nino or La nina years in that they showed sustained warming or cooling exceeding (I think) 5 months. Averaging across the 12 months for a single value therefore can only tell you if its positive or negative SOI, not whether it’s an El nino or La nina year. Therefore, as long you are using averaging across months, you should only refer to positive or negative SOI. You would need to remove the last sentence of Fig 2’s caption and adapt elsewhere where the terms are mentioned, including abstract....

Decision letter (RSOS-211399.R0)

Dear Miss Geeson

The Editors assigned to your paper RSOS-211399 "Environmental influences on breeding biology and pup production in Australian fur seals" have now received comments from reviewers and would like you to revise the paper in accordance with the reviewer comments and any comments from the Editors. Please note this decision does not guarantee eventual acceptance.

We invite you to respond to the comments supplied below and revise your manuscript. Below the referees’ and Editors’ comments (where applicable) we provide additional requirements. Final acceptance of your manuscript is dependent on these requirements being met. We provide guidance below to help you prepare your revision.

Please submit your revised manuscript and required files (see below) no later than 21 days from today's (ie 28-Oct-2021) date. Note: the ScholarOne system will ‘lock’ if submission of the revision is attempted 21 or more days after the deadline. If you do not think you will be able to meet this deadline please contact the editorial office immediately.

Please note article processing charges apply to papers accepted for publication in Royal Society Open Science (<https://royalsocietypublishing.org/rsos/charges>). Charges will also apply to papers transferred to the journal from other Royal Society Publishing journals, as well as papers submitted as part of our collaboration with the Royal Society of Chemistry

(<https://royalsocietypublishing.org/rsos/chemistry>). Fee waivers are available but must be requested when you submit your revision (<https://royalsocietypublishing.org/rsos/waivers>).

on behalf of Dr Ari Friedlaender (Associate Editor) and Pete Smith (Subject Editor)
openscience@royalsociety.org

Associate Editor Comments to Author (Dr Ari Friedlaender):

Comments to the Author:

To the Authors,

I have received feedback from two external reviewers and based on their evaluations am recommending major revisions. The aims and scope of the study are both of interest and done at a high level, however there are two main areas of concern that I believe will require substantial effort to rectify. As noted by the reviewer, the reproductive metrics being used a dependent variables in the modeling work need to be justified and their merit/value must be expanded on to show precisely why each is being used and how each can inform the model results. As well, the reviewer also appropriately notes that the binary use of el nino/la nino is not appropriate and requires attention.

As well, each reviewer provides more minor comments that will require attention moving forward.

Thank you and please let me know if you have any questions.

Ari S. Friedlaender

Reviewer comments to Author:

Reviewer: 1

Comments to the Author(s)

The authors examine environmental factors that could be modulating pup production in the Australian fur seal over multiple decades. They do an admirable job of tackling difficult questions, particularly in terms of their consideration of the environmental inputs to consider and their interpretation and discussion of their results. The mechanisms linking predator reproduction to ocean conditions are complex and their manuscript reflects that. Their analyses and results generally seem robust and well thought through. I would suggest that the authors look at streamlining and focusing the introduction somewhat. I recognize that there are a lot of concepts to introduce, both biologically and oceanographically, but some of the information included there felt extraneous or like it could be presented more concisely. I've provided some specific suggestions and comments here:

Lines 51-80: I understand that a lot of this information is useful background, but I believe it could be restructured and rearranged somewhat to introduce the concepts more concisely and smoothly and to transition between the broad introduction and study objectives.

Lines 51-57: This paragraph in particular feels somewhat dropped in. Additionally, the second sentence is true for income breeding pinniped species, but not really for most capital breeders, especially those on the more extreme side of the capital/income axis. Lastly, there is no real explanation of what an income breeder is in this paragraph, which is fine for the audience that

studies seabirds/marine mammals, but might be needed clarification for broader audiences (especially since income breeding can mean different things for different organisms).

Lines 73-75: This concept feels like it should be integrated with your earlier description of fur seal reproduction. The statement on line 70-72 ("population dynamics of otariid seals...") summarizes the information in this later paragraph as well as it does that previous paragraph

Lines 107-110: These statements about the broader economic/social implications of your study are interesting, but they are not revisited in the discussion/conclusion of your manuscript. Suggest removing.

Line 135: since you only use the term CMR once, and in that same sentence, I suggest not adding it as an acronym.

Line 354-355: wording suggestion "El Niño events cause significant crashes in both the number of pups born and/or high subsequent pup mortality."

Lines 374-377: It might be interesting to know/discuss how this fecundity rate compares to that of other fur seal species, especially following periods of exploitation.

Reviewer: 2

Comments to the Author(s)

The study has shown a lot of interactions between each of four breeding indicators and a lot of environmental predictors. Firstly, I have some concerns with the reliability of some of the breeding parameters, or the support for them. Actual pup production seems fine - it is a well-established indicator. It is acknowledged (148) that it might not accurately reflect trends in breeding population size, but we are not looking at trends in breeding population size we are looking at environmental influences on breeding biology, at an annual scale (with lags). I am a bit unconvinced about the veracity of the fecundity index introduced next (150): the ratio between females and pups in the colony. It supposes that females that abort their fetuses before their full period still attend the breeding colony in breeding season to mate. The reference given to support this is Gibbens et al. 2010 - but I cannot find any evidence in that paper to support this. A fecundity index such as this would probably need to be assessed by looking at its relationship with pregnancy estimation in females sampled before the breeding season, each year. The median pupping date as an indicator of variability supposes a relationship between median pupping date and abortion rate in younger females (which have been shown to pup earlier than older females), but has such a relationship been shown? Please provide support.

Also, are the breeding parameters related to one another as one would expect? If they are not, this might also put the meaningfulness of some of the relationships with environmental predictors into doubt. For example, the relationship between SOI and pup production suggests that positive SOI has a positive influence on pup production in the same year. But the positive relationship between SOI and median pupping date is taken to mean that positive SOI in one year has a negative influence on foraging in the next year, causing later median pupping date. So it should be expected that later median pupping is associated with lower pup production (same year). Is this the case? Likewise, the relationship of these breeding variables with fecundity, and pupping period, should be established. Is the median date actually positively related to pupping period, as expected (311)? No real evidence is cited to support it. So lack of support for some of these breeding parameters seems a weakness of this study; some seem to be more or less assumed based on reasoning, which makes the study seem highly speculative. I would like to see stronger support for the parameters, or at least greater acknowledgment of their potential limitations and the implications for results and interpretation. And I would like to see how the different breeding indicators used here relate to each other. It feels as though this should be an exercise at the outset,

and based on the correlations, or lack thereof between breeding parameters, and/or solid support for them, it might be appropriate to not select one or more of them for modelling with environmental parameters.

More specific comments:

52-57: Not sure that ice (sea ice) should be seen as terrestrial. Suggest changing the sentence as follows: "Consequently, marine predators that breed on land or ice (e.g. pinnipeds, seabirds) are susceptible to local environmental perturbations as their foraging range is limited by access to reliable oceanic prey sources near potential sites for mating and parturition [10]." (However, susceptibility to local environmental perturbations is much more relevant to income breeders than capital breeders, whose foraging distribution may be much further removed from breeding habitat. Perhaps it should be "Consequently, income breeding marine predators....", with modification of the next sentence)

87: Is this the average body mass? Please state...

L89: It might be useful to specifically also contrast the slow recovery of the AuFS with the recovery of its conspecific, A.P. pusillus

100: For the statement: "There is presently little information on how environmental variation in the region influences marine prey species and the breeding biology of predators that depend on them" - it's unclear why studies from other regions are cited. E.g. Benguela, South America

216: write out "Two hundred" to start the sentence

255, Fig 2 and elsewhere: based on this, every year in the study is either an El Nino or La Nina year, which is not true, only some of these years would have El nino or La nina years in that they showed sustained warming or cooling exceeding (I think) 5 months. Averaging across the 12 months for a single value therefore can only tell you if its positive or negative SOI, not whether it's an El nino or La nina year. Therefore, as long you are using averaging across months, you should only refer to positive or negative SOI. You would need to remove the last sentence of Fig 2's caption and adapt elsewhere where the terms are mentioned, including abstract....

===PREPARING YOUR MANUSCRIPT===

If you have been asked to revise the written English in your submission as a condition of publication, you must do so, and you are expected to provide evidence that you have received

language editing support. The journal would prefer that you use a professional language editing service and provide a certificate of editing, but a signed letter from a colleague who is a native speaker of English is acceptable. Note the journal has arranged a number of discounts for authors using professional language editing services (<https://royalsociety.org/journals/authors/benefits/language-editing/>).

===PREPARING YOUR REVISION IN SCHOLARONE===

<https://royalsociety.org/journals/authors/author-guidelines/#supplementary-material> to

include a suitable title and informative caption. An example of appropriate titling and captioning may be found at https://figshare.com/articles/Table_S2_from_Is_there_a_trade-off_between_peak_performance_and_performance_breadth_across_temperatures_for_aerobic_sc_ope_in_teleost_fishes_/3843624.

Author's Response to Decision Letter for (RSOS-211399.R0)

See Appendix A.

RSOS-211399.R1 (Revision)

Review form: Reviewer 2

Is the manuscript scientifically sound in its present form?

Yes

Are the interpretations and conclusions justified by the results?

Yes

Is the language acceptable?

Yes

Do you have any ethical concerns with this paper?

No

Have you any concerns about statistical analyses in this paper?

No

Recommendation?

Accept with minor revision (please list in comments)

Comments to the Author(s)

The line numbers referred to in the author responses do not always align with the correct place in the revised (track changes) pdf version I received. However, I was able to satisfy myself that the responses to the more specific comments of Reviewer 2 (myself) and those of Reviewer 1 are adequate. Some of the responses to the more general comments of Reviewer 2 are a bit more difficult to follow and in some instances not quite convincing. See below....

1st general comment Reviewer 2 ("The study has shown..."):

While the authors' response does not seem to refer to any specific place in the text, I think that the following change at line 162-163 is probably relevant: "Previous studies have shown that adult females that abort their foetus still attend the colony during the breeding season to mate [60]". They refer only to one study however, by Gibbens et al. Therefore, it should rather say "A previous study has shown...". However, looking at that study, nowhere there does it say that

adult females that have aborted their foetus are present in the breeding season to mate. Instead, the authors (this study) have inferred this from retrospectively looking at the Gibbens et al. data. This is clear in their response to the reviewer: “We have, however, determined from the original Gibbens et al. 2010 data....”. The authors of this study need to be clear about this and need to be cautious in their wording, because future studies may read this and cite it as fact, although there is not actually direct evidence, even if it may seem intuitive to the authors (e.g. the Gibbens et al paper acknowledge that early pup mortality may contribute to the numbers of females sighted consecutively without pups during the breeding season). They should rather word it something along these lines: “from the Gibbens et al. 2010 study, it may be inferred that adult females that abort their foetus [are likely to / may / probably] attend....”

2nd general comment Reviewer 2 (“The median pupping date...”):

I think that the response refers to changes made at lines 333-335 in my version. I’m not sure that these changes address my concern and from the response to reviewer, the relationship still seems quite tenuous. At any rate, the text adjustment would be improved if it said: “may indicate poor foraging conditions in the preceding winter, with greater impacts on younger than on older pregnant females.” Further up (lines 329-330), I suggest “Previous studies have shown that larger (presumably older) female otariids give birth later in the pupping season than smaller (presumably younger) females [52, 59, 95, 96], and that the latter are more likely to abort pregnancy.....”

3rd general comment Reviewer 2 (“Also, are the breeding parameters...”):

The authors respond that “While one might expect these breeding parameters to be related, there is no a priori reason to support this.” But to expect it in the first place must be based on some a priori reason....their next sentence: “We investigated these parameters expecting them to be related...”. I don’t agree with the authors’ reasoning that my contention that it “should be expected that later median pupping is associated with lower pup production (same year)” would only likely apply if the breeding population size was static (quoting the authors: “With inter-annual fluctuations in pup production (and recruitment and likely adult survival) the available adult female breeding population is unlikely to be static.”). The population does not have to be static to suppose that after 16 years, one might notice a relationship between pup production and median pupping date, if both are meaningful parameters and both are influenced by environmental variables, even if the relationship may not be direct. Just as, the relationship referred to in the introduced text (lines 351-359 in my version), namely between cohort size and first year survival rate (Arnould unpublished data), also would not suppose a static breeding population size.

Having said this, the introduced text probably is sufficient to support using the parameter in question (median pupping date). Also, the second sentence of the Results seems to indicate that the breeding parameters are all significantly correlated anyway (see also a comment of mine further down).

4th general comment Reviewer 2 (“Likewise, the relationship...”):

Similar to above. It’s a given that there will be inter-annual variation in the numbers of females attending the colony, and that survivorship, recruitment etc may affect this. Nevertheless, after a long time series one may expect to relationships developing between breeding parameters, where expected, such as between median pupping and pup production, or with fecundity index etc. The authors say: “It is the lack of strong relationships between these variables that leads us to feel that the environmental influences on all aforementioned breeding parameters are important in the comprehensive understanding of the breeding biology of Australian fur seals”. On the contrary, I think that strong relationship between some of the parameters would attest more clearly to environmental influence, because, as they have been defined, pup production, fecundity, median pupping and length of pupping should all really react in expected ways to a common environmental fluctuation. The fact that they may not can be taken to mean that the

relationships are more complex than expected, as the authors say somewhere, but the possibility remains that one or more of the parameters may not be that reliable, in particular those that are built on a number of assumptions (I suppose I am most concerned with the fecundity index). I'm okay with the authors using all 4 of the breeding parameters, but there needs to be some admission of the potential limitations of parameters, in view of the assumptions that they are built on, and that this may affect their relationships with one another and to environmental variables. Such 'limitations and assumptions' should be standard for a paper that used indices such as these, but I am not seeing them.

5th general comment Reviewer 2 ("No real evidence..."):

Similar to above. The authors add support for median pupping date and pupping period (lines 142-146 in my version). I still don't think that they address potential limitations (especially regarding fecundity index), which was one of the points I made in my initial review. Introducing some discussion on the newly inserted results reporting on strength of correlations between breeding parameters, might present a sensible place to discuss potential limitations.

Regarding these results, the second line of Results says: "The timing of breeding parameters investigated in the present were significantly, but weakly correlated ($r^2 < 0.28$, $P > 0.02$)". I think it should be "present study". If this is a Pearson correlation, as indicated in Methods, why is R squared reported? Also, it would need to be $r^2 = 0.28$, not simply r^2 . Finally, if they were all significantly correlated, why present $P > 0.02$. Should it not be $P < 0.05$?

Also related to this, on lines 417-418 (my version), the following has been inserted:

"Contrastingly, the more abundant conspecific Cape fur seal has a reported fecundity rate of 77.5% [111]". It is very important to qualify (a) that this was measured when the Cape fur seal population was rapidly increasing, (b) that it used a very different approach to the method used here (necropsies of shot animals).

Decision letter (RSOS-211399.R1)

Dear Miss Geeson

On behalf of the Editors, we are pleased to inform you that your Manuscript RSOS-211399.R1 "Environmental influences on breeding biology and pup production in Australian fur seals" has been accepted for publication in Royal Society Open Science subject to minor revision in accordance with the referees' reports. Please find the referees' comments along with any feedback from the Editors below my signature.

Please submit your revised manuscript and required files (see below) no later than 7 days from today's (ie 09-Mar-2022) date. Note: the ScholarOne system will 'lock' if submission of the revision is attempted 7 or more days after the deadline. If you do not think you will be able to meet this deadline please contact the editorial office immediately.

on behalf of Dr Ari Friedlaender (Associate Editor) and Pete Smith (Subject Editor)
openscience@royalsociety.org

Reviewer comments to Author:

Reviewer: 2

Comments to the Author(s)

The line numbers referred to in the author responses do not always align with the correct place in the revised (track changes) pdf version I received. However, I was able to satisfy myself that the responses to the more specific comments of Reviewer 2 (myself) and those of Reviewer 1 are adequate. Some of the responses to the more general comments of Reviewer 2 are a bit more difficult to follow and in some instances not quite convincing. See below....

1st general comment Reviewer 2 ("The study has shown..."):

While the authors' response does not seem to refer to any specific place in the text, I think that the following change at line 162-163 is probably relevant: "Previous studies have shown that adult females that abort their foetus still attend the colony during the breeding season to mate [60]". They refer only to one study however, by Gibbens et al. Therefore, it should rather say "A previous study has shown...". However, looking at that study, nowhere there does it say that adult females that have aborted their foetus are present in the breeding season to mate. Instead, the authors (this study) have inferred this from retrospectively looking at the Gibbens et al. data. This is clear in their response to the reviewer: "We have, however, determined from the original Gibbens et al. 2010 data....". The authors of this study need to be clear about this and need to be cautious in their wording, because future studies may read this and cite it as fact, although there is not actually direct evidence, even if it may seem intuitive to the authors (e.g. the Gibbens et al paper acknowledge that early pup mortality may contribute to the numbers of females sighted consecutively without pups during the breeding season). They should rather word it something along these lines: "from the Gibbens et al. 2010 study, it may be inferred that adult females that abort their foetus [are likely to / may / probably] attend...."

2nd general comment Reviewer 2 ("The median pupping date..."):

I think that the response refers to changes made at lines 333-335 in my version. I'm not sure that these changes address my concern and from the response to reviewer, the relationship still seems quite tenuous. At any rate, the text adjustment would be improved if it said: "may indicate poor foraging conditions in the preceding winter, with greater impacts on younger than on older pregnant females." Further up (lines 329-330), I suggest "Previous studies have shown that larger (presumably older) female otariids give birth later in the pupping season than smaller (presumably younger) females [52, 59, 95, 96], and that the latter are more likely to abort pregnancy....."

3rd general comment Reviewer 2 (“Also, are the breeding parameters...”):

The authors respond that “While one might expect these breeding parameters to be related, there is no a priori reason to support this.” But to expect it in the first place must be based on some a priori reason....their next sentence: “We investigated these parameters expecting them to be related...”. I don’t agree with the authors’ reasoning that my contention that it “should be expected that later median pupping is associated with lower pup production (same year)” would only likely apply if the breeding population size was static (quoting the authors: “With inter-annual fluctuations in pup production (and recruitment and likely adult survival) the available adult female breeding population is unlikely to be static.”). The population does not have to be static to suppose that after 16 years, one might notice a relationship between pup production and median pupping date, if both are meaningful parameters and both are influenced by environmental variables, even if the relationship may not be direct. Just as, the relationship referred to in the introduced text (lines 351-359 in my version), namely between cohort size and first year survival rate (Arnould unpublished data), also would not suppose a static breeding population size.

Having said this, the introduced text probably is sufficient to support using the parameter in question (median pupping date). Also, the second sentence of the Results seems to indicate that the breeding parameters are all significantly correlated anyway (see also a comment of mine further down).

4th general comment Reviewer 2 (“Likewise, the relationship...”):

Similar to above. It’s a given that there will be inter-annual variation in the numbers of females attending the colony, and that survivorship, recruitment etc may affect this. Nevertheless, after a long time series one may expect to relationships developing between breeding parameters, where expected, such as between median pupping and pup production, or with fecundity index etc.

The authors say: “It is the lack of strong relationships between these variables that leads us to feel that the environmental influences on all aforementioned breeding parameters are important in the comprehensive understanding of the breeding biology of Australian fur seals”. On the contrary, I think that strong relationship between some of the parameters would attest more clearly to environmental influence, because, as they have been defined, pup production, fecundity, median pupping and length of pupping should all really react in expected ways to a common environmental fluctuation. The fact that they may not can be taken to mean that the relationships are more complex than expected, as the authors say somewhere, but the possibility remains that one or more of the parameters may not be that reliable, in particular those that are built on a number of assumptions (I suppose I am most concerned with the fecundity index). I’m okay with the authors using all 4 of the breeding parameters, but there needs to be some admission of the potential limitations of parameters, in view of the assumptions that they are built on, and that this may affect their relationships with one another and to environmental variables. Such ‘limitations and assumptions’ should be standard for a paper that used indices such as these, but I am not seeing them.

5th general comment Reviewer 2 (“No real evidence...”):

Similar to above. The authors add support for median pupping date and pupping period (lines 142-146 in my version). I still don’t think that they address potential limitations (especially regarding fecundity index), which was one of the points I made in my initial review. Introducing some discussion on the newly inserted results reporting on strength of correlations between breeding parameters, might present a sensible place to discuss potential limitations.

Regarding these results, the second line of Results says: “The timing of breeding parameters investigated in the present were significantly, but weakly correlated ($r^2 < 0.28$, $P > 0.02$)”. I think it should be “present study”. If this is a Pearson correlation, as indicated in Methods, why is R squared reported? Also, it would need to be ≤ 0.28 , not simply $<$. Finally, if they were all

significantly correlated, why present $P > 0.02$. Should it not be $P < 0.05$?

Also related to this, on lines 417-418 (my version), the following has been inserted: "Contrastingly, the more abundant conspecific Cape fur seal has a reported fecundity rate of 77.5% [111]". It is very important to qualify (a) that this was measured when the Cape fur seal population was rapidly increasing, (b) that it used a very different approach to the method used here (necropsies of shot animals).

===PREPARING YOUR MANUSCRIPT===

one version should clearly identify all the changes that have been made (for instance, in coloured highlight, in bold text, or tracked changes);

===PREPARING YOUR REVISION IN SCHOLARONE===

-- If you are requesting an article processing charge waiver, you must select the relevant waiver option (if requesting a discretionary waiver, the form should have been uploaded, see 'File upload' above).

-- If you have uploaded any electronic supplementary (ESM) files, please ensure you follow the guidance at <https://royalsociety.org/journals/authors/author-guidelines/#supplementary-material> to include a suitable title and informative caption. An example of appropriate titling and captioning may be found at https://figshare.com/articles/Table_S2_from_Is_there_a_trade-off_between_peak_performance_and_performance_breadth_across_temperatures_for_aerobic_scope_in_teleost_fishes_/3843624.

Author's Response to Decision Letter for (RSOS-211399.R1)

Appendix B.

Decision letter (RSOS-211399.R2)

Dear Miss Geeson,

I am pleased to inform you that your manuscript entitled "Environmental influences on breeding biology and pup production in Australian fur seals" is now accepted for publication in Royal Society Open Science.

on behalf of Dr Ari Friedlaender (Associate Editor) and Pete Smith (Subject Editor)
openscience@royalsociety.org

Appendix A

Dear Dr Ari Friedlaender,

We thank the Associate Editor and reviewers for their detailed and constructive feedback on the manuscript “Environmental influences on breeding biology and pup production in Australian fur seals”. We have integrated these suggestions, where appropriate, into the manuscript and feel this has greatly strengthened and clarified the manuscript.

The main changes to the manuscript include:

- 1) We have added clarification and further support for the breeding parameters used in the study in determining the environmental influences on breeding biology and pup production in Australian fur seals. We have improved contextualisation and provided greater support from the existing literature. Additionally, we have included further justification for the inclusion of each parameter due to the independence of each parameter, and thus relevance in a comprehensive overview of Australian fur seal breeding biology.
- 2) We have referred to SOI conditions and trends rather than recognised El Niño and La Niña events to ensure accuracy and clarity of the presentation of our findings.

Detailed responses to the reviewer comments are included below. The manuscript has been supplied with track changes and as a finalised clean copy without track changes. Please note that the page and line numbers given in the point-by-point response are based on the clean version.

We look forward to hearing the outcome of the next round of reviews.

Kind regards,

Johanna Geeson

Reviewer: 1

The authors examine environmental factors that could be modulating pup production in the Australian fur seal over multiple decades. They do an admirable job of tackling difficult questions, particularly in terms of their consideration of the environmental inputs to consider and their interpretation and discussion of their results. The mechanisms linking predator reproduction to ocean conditions are complex and their manuscript reflects that. Their analyses and results generally seem robust and well thought through. I would suggest that the authors look at streamlining and focusing the introduction somewhat. I recognize that there are a lot of concepts to introduce, both biologically and oceanographically, but some of the information included there felt extraneous or like it could be presented more concisely. I've provided some specific suggestions and comments here:

Please see the responses to the above concerns in the detailed comments below.

Lines 51-80: I understand that a lot of this information is useful background, but I believe it could be restructured and rearranged somewhat to introduce the concepts more concisely and smoothly and to transition between the broad introduction and study objectives.

We have modified these two paragraphs to streamline the concepts and transitions between them [Lines: 51-61].

Lines 51-57: This paragraph in particular feels somewhat dropped in. Additionally, the second sentence is true for income breeding pinniped species, but not really for most capital breeders, especially those on the more extreme side of the capital/income axis. Lastly, there is no real explanation of what an income breeder is in this paragraph, which is fine for the audience that studies seabirds/marine mammals, but might be needed clarification for broader audiences (especially since income breeding can mean different things for different organisms).

Good point. We have included a definition of income-breeding as it relates to seabirds/marine mammals in this paragraph and rephrased it somewhat to provide a smoother transition between concepts [Lines: 52-55].

Lines 73-75: This concept feels like it should be integrated with your earlier description of fur seal reproduction. The statement on line 70-72 ("population dynamics of otariid seals....") summarizes the information in this later paragraph as well as it does that previous paragraph

While we understand why this was suggested, we believe the second statement provides required context for the study, emphasising specific limitations on female otariids in conjunction with the earlier description relating more broadly to income-breeding species. We have altered both sections to be more specific and less repetitive [Lines: 59-61 & 71-75].

Lines 107-110: These statements about the broader economic/social implications of your study are interesting, but they are not revisited in the discussion/conclusion of your manuscript. Suggest removing. **Good point. These have been removed [Line: 112].**

Line 135: since you only use the term CMR once, and in that same sentence, I suggest not adding it as an acronym.
Amended [Line: 141].

Line 354-355: wording suggestion “El Niño events cause significant crashes in both the number of pups born and/or high subsequent pup mortality.”
Nice suggestion. The text has been edited to the suggested wording [Lines: 370-371].

Lines 374-377: It might be interesting to know/discuss how this fecundity rate compares to that of other fur seal species, especially following periods of exploitation.
We agree and hence have added comparison to the fecundity rates of other fur seals [Line: 389].

Reviewer: 2

The study has shown a lot of interactions between each of four breeding indicators and a lot of environmental predictors. Firstly, I have some concerns with the reliability of some of the breeding parameters, or the support for them. Actual pup production seems fine - it is a well-established indicator. It is acknowledged (**148**) that it might not accurately reflect trends in breeding population size, but we are not looking at trends in breeding population size we are looking at environmental influences on breeding biology, at an annual scale (with lags). I am a bit unconvinced about the veracity of the fecundity index introduced next (**150**): the ratio between females and pups in the colony. It supposes that females that abort their fetuses before their full period still attend the breeding colony in breeding season to mate. The reference given to support this is Gibbens et al. 2010 - but I cannot find any evidence in that paper to support this. A fecundity index such as this would probably need to be assessed by looking at its relationship with pregnancy estimation in females sampled before the breeding season, each year.

As referenced in the paragraph of the methods starting with “As fur seals are long-lived mammals...”[Line: 151] and in the discussion starting with “The average Fecundity Index observed...”[Line: 384], the reviewer’s suggestion on how the fecundity index needs to be assessed is how the fecundity was estimated in Gibbens et al. 2010. As stated in the Methods section of the Gibbens et al. 2010 paper: “birth status was estimated only for those females for which ≥ 2 consistent resights were recorded (i.e., observed ≥ 2 times with young or ≥ 2 times alone). Additionally, resights were required to occur ≥ 2 days apart so that they were likely to have been separated by a maternal foraging trip and, therefore, represent discrete attendance events (Arnould and Hindell 2001). Females not meeting these criteria were assigned unknown birth status”. The pupping rate reported by Gibbens et al. 2010 was calculated simply as a

proportion of females sampled for pregnancy during the previous winter that pupped. As only 81-85% of females sampled for pregnancy were pregnant, some of the non-pupping females observed during the breeding season may not have been pregnant during the winter. We have, however, determined from the original Gibbens et al. 2010 data that of the females that were deemed pregnant during the winter that were resighted ≥ 2 times during the breeding season, only 57% pupped. Hence, these findings indicated that the majority of females that had aborted a foetus still attended the colony during the breeding season, presumably for mating (as indicated by the pregnancy rates in the following winter observed by Gibbens et al.).

We fully acknowledge that the Fecundity Index developed is not perfect, which is why it is referred to as an index rather than an estimation of true fecundity of the species. We explored this index in trying to understand whether pupping rates varied interannually (the Gibbens et al 2010 paper was limited to 3 years, whereas this data set expanded the record to 16 years) and what factors might influence it. The surprising finding that measures of oceanic productivity were not positively related to the fecundity index raises interesting possibilities that we are currently exploring i.e., diseases that promote abortions may be more influenced by ambient weather conditions than prey availability.

The median pupping date as an indicator of variability supposes a relationship between median pupping date and abortion rate in younger females (which have been shown to pup earlier than older females), but has such a relationship been shown? Please provide support.

We cannot and do not suggest to show a direct relationship between the median pupping date and abortion rate in females. However, we are offering this as a potential explanation. Lourie et al. 2014 showed a relationship between larger (and presumably older, Guinet et al 1998) females giving birth later in the season, and Gibbens et al. 2009 found that Australian fur seals produce less offspring in years of low body condition, while Guinet et al. 1998 found low body condition and, thus, poor foraging conditions to be linked to higher rates of abortion in Cape fur seals. Additionally, Gibbens et al. 2010 found body size and body condition to increase with age in Australian fur seals. The text has been adjusted to clarify the above mentioned relationships [Lines: 315-317].

Also, are the breeding parameters related to one another as one would expect? If they are not, this might also put the meaningfulness of some of the relationships with environmental predictors into doubt. For example, the relationship between SOI and pup production suggests that positive SOI has a positive influence on pup production in the same year. But the positive relationship between SOI and median pupping date is taken to mean that positive SOI in one year has a negative influence on foraging in the next year, causing later median pupping date. So it should be expected that later median pupping is associated with lower pup production (same year). Is this the case?

While one might expect these breeding parameters to be related, there is no *a priori* reason to support this. We investigated these parameters expecting them to be related but found complex interactions between pup production in one year and timing of breeding in the next in relation to environmental influences (and with lags). While we do not have all the answers, we have suggested a potential scenario related to the influence of density-dependence in pup numbers supported by unpublished findings from demographic studies at a nearby colony [Lines: 348-357]. Furthermore, the reviewer's contention that "should be expected that later median pupping is associated with lower pup production (same year)" would only likely apply if the breeding population size was static. With inter-annual fluctuations in pup production (and recruitment and likely adult survival) the available adult female breeding population is unlikely to be static.

Likewise, the relationship of these breeding variables with fecundity, and pupping period, should be established.

While relationships between some of the variables, such as the fecundity index and pup production, may have been expected, inter-annual variation in the population of adult females that attend the colony (due to variation in recruitment or survivorship etc.) may affect these relationships. It is the lack of strong relationships between these variables that leads us to feel that the environmental influences on all aforementioned breeding parameters are important in the comprehensive understanding of the breeding biology of Australian fur seals.

No real evidence is cited to support it. So lack of support for some of these breeding parameters seems a weakness of this study; some seem to be more or less assumed based on reasoning, which makes the study seem highly speculative. I would like to see stronger support for the parameters, or at least greater acknowledgment of their potential limitations and the implications for results and interpretation. And I would like to see how the different breeding indicators used here relate to each other. It feels as though this should be an exercise at the outset, and based on the correlations, or lack thereof between breeding parameters, and/or solid support for them, it might be appropriate to not select one or more of them for modelling with environmental parameters.

On this suggestion we have increased the support of these breeding parameters in the text, particularly the relevance and importance of the median pupping date and pupping period [Lines: 137-138]. While many studies looking at the influence and effects of timing of breeding are correlative rather than causative, it has been determined that there is a functional relationship between timing of breeding and nutrition in species including Australian fur seals (Gibbens & Arnould 2009), South American sea lions (Soto et al. 2004), Antarctic fur seals (Boyd 1996), northern fur seals (Trites 1992) and Steller sea lions (Pitcher et al. 2001). Hence, we feel justified in retaining all four breeding parameters used in the study.

More specific comments:

Not sure that ice (sea ice) should be seen as terrestrial. Suggest changing the sentence as follows:
“Consequently, marine predators that breed on land or ice (e.g. pinnipeds, seabirds) are susceptible to local environmental perturbations as their foraging range is limited by access to reliable oceanic prey sources near potential sites for mating and parturition [10].” (However, susceptibility to local environmental perturbations is much more relevant to income breeders than capital breeders, whose foraging distribution may be much further removed from breeding habitat. Perhaps it should be “Consequently, income breeding marine predators....”, with modification of the next sentence)

Good point. The statement has been rephrased, separating sea ice from terrestrial land and putting the emphasis on the income-breeding aspect of these marine predators rather than terrestrial-breeding.

[Lines: 52-55]

87: Is this the average body mass? Please state...

Yes, the text has been amended for clarity [Line: 90].

L89: It might be useful to specifically also contrast the slow recovery of the AuFS with the recovery of its conspecific, *A.P. pusillus*

Nice suggestion. This has been included in the text [Lines: 94-95].

100: For the statement: “There is presently little information on how environmental variation in the region influences marine prey species and the breeding biology of predators that depend on them” – it’s unclear why studies from other regions are cited. E.g. Benguela, South America

The sentence has been restructured to clarify that we were referring to general regional environmental variation rather than specific environmental variation within Bass Strait [Line: 106].

216: write out “Two hundred” to start the sentence

Amended [Line: 222].

255, Fig 2 and elsewhere: based on this, every year in the study is either an El Nino or La Nina year, which is not true, only some of these years would have El nino or La nina years in that they showed sustained warming or cooling exceeding (I think) 5 months. Averaging across the 12 months for a single value therefore can only tell you if its positive or negative SOI, not whether it’s an El nino or La nina year. Therefore, as long you are using averaging across months, you should only refer to positive or negative SOI. You would need to remove the last sentence of Fig 2’s caption and adapt elsewhere where the terms are mentioned, including abstract....

We have modified this in the text and in Figure 2 to reflect that we are looking at SOI conditions which relate to but are independent from recognised El Niño and La Niña events [Lines: 33-34, 272, 286-287, 387, 390-394, 405-407, 451, 829-830].

Appendix B

Dear Dr Ari Friedlaender,

We thank the Associate Editor and reviewers for their comments on the manuscript "Environmental influences on breeding biology and pup production in Australian fur seals". We have integrated the suggested changes into the manuscript as requested. Specifically, further clarification on the limitations and assumptions of the breeding indices used, and their potential implications, has been included throughout the text.

Detailed responses to the reviewer comments are included below. A copy of the manuscript has been supplied with areas of changes in the text highlighted and as a finalised clean copy without highlighted text.

Kind regards,

Johanna Geeson

1st general comment Reviewer 2:

While the authors' response does not seem to refer to any specific place in the text, I think that the following change at line 162-163 is probably relevant: "Previous studies have shown that adult females that abort their foetus still attend the colony during the breeding season to mate [60]".

They refer only to one study however, by Gibbens et al. Therefore, it should rather say "A previous study has shown...". However, looking at that study, nowhere there does it say that adult females that have aborted their foetus are present in the breeding season to mate. Instead, the authors (this study) have inferred this from retrospectively looking at the Gibbens et al. data. This is clear in their response to the reviewer: "We have, however, determined from the original Gibbens et al. 2010 data....". The authors of this study need to be clear about this and need to be cautious in their wording, because future studies may read this and cite it as fact, although there is not actually direct evidence, even if it may seem intuitive to the authors (e.g. the Gibbens et al paper acknowledge that early pup mortality may contribute to the numbers of females sighted consecutively without pups during the breeding season). They should rather word it something along these lines: "from the Gibbens et al. 2010 study, it may be inferred that adult females that abort their foetus [are likely to / may / probably] attend...."

We have clarified the text by paraphrasing the reviewer's suggestion and amending the text [lines: 153-155].

2nd general comment Reviewer 2 ("The median pupping date..."):

I think that the response refers to changes made at lines 333-335 in my version. I'm not sure that these changes address my concern and from the response to reviewer, the relationship still seems quite tenuous. At any rate, the text adjustment would be improved if it said: "may indicate poor foraging conditions in the preceding winter, with greater impacts on younger than on older pregnant females." Further up (lines 329-330), I suggest "Previous studies have shown that larger (presumably older) female otariids give birth later in the pupping season than smaller (presumably younger) females [52, 59, 95, 96], and that the latter are more likely to abort pregnancy...."

The text has been amended as recommended [lines: 321-326].

3rd general comment Reviewer 2 ("Also, are the breeding parameters..."):

The authors respond that "While one might expect these breeding parameters to be related, there is no a priori reason to support this." But to expect it in the first place must be based on some a priori reason....their next sentence: "We investigated these parameters expecting them to be related...". I don't agree with the authors' reasoning that my contention that it "should be expected that later median pupping is associated with lower pup production (same year)" would only likely apply if the breeding population size was static (quoting the authors: "With inter-annual fluctuations in pup production (and

recruitment and likely adult survival) the available adult female breeding population is unlikely to be static.”). The population does not have to be static to suppose that after 16 years, one might notice a relationship between pup production and median pupping date, if both are meaningful parameters and both are influenced by environmental variables, even if the relationship may not be direct. Just as, the relationship referred to in the introduced text (lines 351-359 in my version), namely between cohort size and first year survival rate (Arnould unpublished data), also would not suppose a static breeding population size.

Having said this, the introduced text probably is sufficient to support using the parameter in question (median pupping date). Also, the second sentence of the Results seems to indicate that the breeding parameters are all significantly correlated anyway (see also a comment of mine further down).

We understand the reviewer's frustrations but note that they believe the previously introduced text is sufficient. As such, we have not amended the text further.

4th general comment Reviewer 2 (“Likewise, the relationship...”):

Similar to above. It's a given that there will be inter-annual variation in the numbers of females attending the colony, and that survivorship, recruitment etc may affect this. Nevertheless, after a long time series one may expect to relationships developing between breeding parameters, where expected, such as between median pupping and pup production, or with fecundity index etc. The authors say: “It is the lack of strong relationships between these variables that leads us to feel that the environmental influences on all aforementioned breeding parameters are important in the comprehensive understanding of the breeding biology of Australian fur seals”. On the contrary, I think that strong relationship between some of the parameters would attest more clearly to environmental influence, because, as they have been defined, pup production, fecundity, median pupping and length of pupping should all really react in expected ways to a common environmental fluctuation. The fact that they may not can be taken to mean that the relationships are more complex than expected, as the authors say somewhere, but the possibility remains that one or more of the parameters may not be that reliable, in particular those that are built on a number of assumptions (I suppose I am most concerned with the fecundity index). I'm okay with the authors using all 4 of the breeding parameters, but there needs to be some admission of the potential limitations of parameters, in view of the assumptions that they are built on, and that this may affect their relationships with one another and to environmental variables. Such 'limitations and assumptions' should be standard for a paper that used indices such as these, but I am not seeing them.

On this suggestion we have included text to address the potential limitations and assumptions of the parameters used in both the methods and discussion sections [lines: 158-160 and 304-307].

5th general comment Reviewer 2 (“No real evidence...”):

Similar to above. The authors add support for median pupping date and pupping period (lines 142-146 in my version). I still don't think that they address potential limitations (especially regarding fecundity index), which was one of the points I made in my initial review. Introducing some discussion on the newly inserted results reporting on strength of correlations between breeding parameters, might present a sensible place to discuss potential limitations.

Regarding these results, the second line of Results says: "The timing of breeding parameters investigated in the present were significantly, but weakly correlated ($r^2 < 0.28$, $P > 0.02$)". I think it should be "present study". If this is a Pearson correlation, as indicated in Methods, why is R squared reported? Also, it would need to be ≤ 0.28 , not simply $<$. Finally, if they were all significantly correlated, why present $P > 0.02$. Should it not be $P < 0.05$?

Also related to this, on lines 417-418 (my version), the following has been inserted: "Contrastingly, the more abundant conspecific Cape fur seal has a reported fecundity rate of 77.5% [111]". It is very important to qualify (a) that this was measured when the Cape fur seal population was rapidly increasing, (b) that it used a very different approach to the method used here (necropsies of shot animals).

We have amended the text as recommended in regards to the note of 'present study' and the p-value suggestions [lines: 241-243 and 270-271]. We used r^2 as a means to express the variability in the dependent variable explained by the independent variable, rather than just the correlation with r

Additionally, the text has been clarified to explain the different methodology in the Cape fur seal fecundity data and that the data was collected when the population was increasing [lines: 394-396].